# Frank-Wolfe-based Algorithms for Approximating Tyler's M-estimator

**Lior Danon**
Technion - Israel Institute of Technology
Haifa, Israel 3200003
liordanon@campus.technion.ac.il

**Dan Garber**
Technion - Israel Institute of Technology
Haifa, Israel 3200003
dangar@technion.ac.il

## Abstract

Tyler's M-estimator is a well known procedure for robust and heavy-tailed covariance estimation. Tyler himself suggested an iterative fixed-point algorithm for computing his estimator however, it requires super-linear (in the size of the data) runtime per iteration, which maybe prohibitive in large scale. In this work we propose, to the best of our knowledge, the first Frank-Wolfe-based algorithms for computing Tyler's estimator. One variant uses standard Frank-Wolfe steps, the second also considers *away-steps* (AFW), and the third is a *geodesic* version of AFW (GAFW). AFW provably requires, up to a log factor, only linear time per iteration, while GAFW runs in linear time (up to a log factor) in a large $n$ (number of data-points) regime. All three variants are shown to provably converge to the optimal solution with sublinear rate, under standard assumptions, despite the fact that the underlying optimization problem is not convex nor smooth. Under an additional fairly mild assumption, that holds with probability 1 when the (normalized) data-points are i.i.d. samples from a continuous distribution supported on the entire unit sphere, AFW and GAFW are proved to converge with linear rates. Importantly, all three variants are parameter-free and use adaptive step-sizes.

## 1 Introduction

Given $n$ data-points in $\mathbb{R}^p$, $\mathbf{x}_1, \ldots, \mathbf{x}_n \in \mathbb{R}^p$ different from zero, Tyler's M-estimator (TME), originally proposed by Tyler in his seminal work [27], is a $p \times p$ positive definite matrix $\mathbf{Q}^*$ which satisfies:

$$\frac{p}{n} \sum_{i=1}^{n} \frac{\mathbf{x}_i \mathbf{x}_i^\top}{\mathbf{x}_i^\top \mathbf{Q}^{*-1} \mathbf{x}_i} = \mathbf{Q}^*. \tag{1}$$

While the TME is not guaranteed to exist for any set of points $\mathbf{x}_1, \ldots, \mathbf{x}_n$, it is well known that under the following assumption proposed in [27], it does.

**Assumption 1.** *The data $\{\mathbf{x}_1, \ldots, \mathbf{x}_n\} \subset \mathbb{R}^p$ satisfies that $\mathbf{x}_i \neq 0$ for all $i = 1, \ldots, n$, and that for any proper subspace $\mathcal{L} \subset \mathbb{R}^p$, denoting by $N(\mathcal{L})$ the number of data-points lying in $\mathcal{L}$, it holds that $n > \frac{N(\mathcal{L})}{\dim(\mathcal{L})} p$.*

The TME is a well known distribution-free robust covariance estimator. It is also a maximum likelihood estimator for the shape matrix of angular and compound Gaussian distributions, and often used for estimation of heavy-tailed distributions [28]. Tyler's M-estimator has received notable interest within the statistics and signal processing communities, see for instance the excellent survey [28] (including the many important references therein), as well as the recent works [30, 11, 29, 25, 6, 16, 20, 24, 2] which provide additional analysis results, related estimators, as well as structured

versions of Tyler's estimator (e.g., convex constraints, low-rank structure, sparse structure, and more), to name only a few.

It is also well known that when the TME exists, it is given by the optimal solution to the following optimization problem:

$$\min_{\mathbf{Q}\in\mathbb{S}^p}\left\{f(\mathbf{Q}) := \frac{p}{n}\sum_{i=1}^{n}\log(\mathbf{x}_i^\top\mathbf{Q}^{-1}\mathbf{x}_i) + \log\det(\mathbf{Q})\right\} \quad \text{s.t.} \quad \mathbf{Q}\succ 0, \mathrm{Tr}(\mathbf{Q}) = p, \qquad (2)$$

where $\mathbb{S}^p$ denotes the set of real-valued symmetric $p\times p$ matrices, and $\mathbf{Q}\succ 0$ ($\mathbf{Q}\succeq 0$) denotes that $\mathbf{Q}$ is positive definite (positive semidefinite).

Indeed the connection between (1) and (2) becomes more apparent when examining the gradient given by

$$\nabla f(\mathbf{Q}) = -\frac{p}{n}\sum_{i=1}^{n}\frac{\mathbf{Q}^{-1}\mathbf{x}_i\mathbf{x}_i^\top\mathbf{Q}^{-1}}{\mathbf{x}_i^\top\mathbf{Q}^{-1}\mathbf{x}_i} + \mathbf{Q}^{-1}. \qquad (3)$$

Thus, a matrix $\mathbf{Q}\succ 0$ satisfying (1), also satisfies $\mathbf{Q}\nabla f(\mathbf{Q})\mathbf{Q} = 0$, meaning $\nabla f(\mathbf{Q}) = 0$, and the other way around. In particular, it can be shown that matrices $\mathbf{Q}\succ 0$ for which $\nabla f(\mathbf{Q}) = 0$, are the only stationary points of Problem (2) (i.e., do not have descent directions).

**Theorem 1** (see for instance [28]). *Under Assumption 1, Problem (2) admits a unique optimal solution $\mathbf{Q}^*\succ 0$. This solution is also the unique solution to Eq. (1) with trace equals $p$.*

Throughout this paper we assume that Assumption 1 indeed holds true. Note also that (1) is invariant to scaling of each data-point $\mathbf{x}_i, i = 1,\ldots,n$, and the matrix $\mathbf{Q}^*$. Hence, throughout this paper, as often customary, we assume that $\mathbf{x}_1,\ldots,\mathbf{x}_n$ are normalized to have unit-length, i.e., $\|\mathbf{x}_i\|_2 = 1, i = 1,\ldots,p$, and $\mathbf{Q}^*$ is normalized to have trace equals $p$, i.e., $\mathrm{Tr}(\mathbf{Q}^*) = p$.

The subject of this paper are efficient algorithms for approximating the TME in large-scale, i.e., when both $n, p$ are large. In [27] Tyler proposed a simple algorithm for computing the TME which performs fixed-point iterations, as we define next.

**Definition 1** (Fixed-point iterations for computing the TME). *The FPI method computes the following iterations:*

$$\mathbf{Q}_{t+1} \leftarrow \frac{p}{n}\sum_{i=1}^{n}\frac{\mathbf{x}_i\mathbf{x}_i^\top}{\mathbf{x}_i^\top\mathbf{Q}_t^{-1}\mathbf{x}_i}, \qquad (4)$$

*and the returned solution is given by $\mathbf{Q} = p\frac{\mathbf{Q}_T}{Tr(\mathbf{Q}_T)}$, where $\mathbf{Q}_T$ is the last iterate computed.* [1]

Note that each iteration (4) requires $O(p^3 + np^2)$ runtime: $O(p^3)$ time in order to compute the inverse matrix $\mathbf{Q}_t^{-1}$ from the previous iterate $\mathbf{Q}_t$, and additional $O(np^2)$ time to compute the sum of rank-one matrices. In particular, the runtime is super-linear in the size of the input which is $np$, and thus can be prohibitive when $n, p$ are both very large (note that under Assumption 1 we always have $n > p$).

In [27] Tyler also proved the convergence of the iteration (4) (under Assumption 1) but without a rate. Recently, it was proved in [6] that when the data $\mathbf{x}_1,\ldots,\mathbf{x}_n$ are i.i.d. samples from an elliptical distribution, the FPI method produces a matrix $\widehat{\mathbf{Q}}\succ 0$ with $\mathrm{Tr}(\widehat{\mathbf{Q}}) = p$, such that with high probability, $\|\mathbf{I} - \mathbf{Q}^{*1/2}\widehat{\mathbf{Q}}^{-1}\mathbf{Q}^{*1/2}\|_F \leq \epsilon$, after $O(|\log\det\Sigma| + p + \log(1/\epsilon))$ iterations, for any error tolerance $\epsilon$, where $\mathbf{Q}^*$ is the solution to (1) with trace equals $p$, $\Sigma$ is the shape matrix of the underlying elliptical distribution, and $\|\cdot\|_F$ denotes the Frobenius (Euclidean) norm. The proof of this result is highly involved and relies on deep mathematical concepts such as strong geodesic convexity and quantum expanders. Note that taking into account the runtime of a single iteration (4) and the bound on number of iterations, the total runtime to reach $\epsilon$ error (according to the measure of convergence in [6]) is $\Omega(p^4 + np^3)$.

The goal of this paper is to present new simple and efficient first-order methods for solving Problem (2) which avoid the super-linear runtime barrier of $np^2$, and even the matrix inversion barrier ($O(p^3)$

---

[1]There is a variant in which the normalization of the trace is performed after each iteration and was observed to have similar empirical performance [28].

in practical implementations), required by each iteration of (4). Our algorithms are based on the well known Frank-Wolfe method (aka conditional gradient method) for constrained smooth minimization [5, 15]. To the best of our knowledge this is the first time that the Frank-Wolfe method has been considered for computing the TME. We provide three such variants with the following properties:

- All three variants provably converge to the optimal solution (the one satisfying Eq. (1)) under Assumption 1 with a sublinear rate — $O(1/\epsilon^2)$ iterations for approximation error $\epsilon$.

- Two variants, AFW and GAFW, are also proved to converge with a linear rate (i.e,. $\log(1/\epsilon)$ dependence on the error tolerance $\epsilon$), under an additional fairly mild assumption that holds with probability 1 when the (normalized) data-points are i.i.d. samples from a continuous distribution supported on the entire unit sphere (note that the $O(\log(1/\epsilon))$ rate proved in [6] also holds under an assumption that the data is sampled i.i.d. from an elliptical distribution).

- All three variants are *parameter-free* and do not require any tuning of parameters, and apply adaptive step-sizes (i.e., not fixed beforehand).

- The AFW variant requires only $\tilde{O}(np)$ runtime per iteration which is linear in the size of the data, up to a single logarithmic factor, while another variant GAFW runs in the same time when $n \geq p^2$. This is in contrast to FPI which requires $O(np^2 + p^3)$ time per iteration. In fact, even if processing the data could be ideally distributed / done in parallel, FPI still requires an expensive matrix inversion on each iteration (which practically runs in $O(p^3)$ time), while our AFW variant only requires additional $O(p^2)$ runtime per iteration.

A quick summary of our results is brought in Table 1.

Importantly and quite pleasingly, our derivations and proofs rely mostly on standard arguments for continuous optimization in Euclidean spaces, and we believe that as such, are quite accessible.

Beyond our contribution to efficient algorithms for computing the TME, our work also contributes more broadly to the theory of Frank-Wolfe-type methods, which have received significant interest in recent years in the context of obtaining faster rates for convex and smooth problems (e.g., [10, 19, 9, 7, 1]), dealing with nonsmooth objectives (e.g., [26, 4, 3]), and dealing with non-convex objectives (e.g., [18]). While Problem (2) is nonsmooth and nonconvex, our Frank-Wolfe variants, with our tailored-designed adaptive step-sizes, are guaranteed to converge to the global minimum, and this is the first result of its kind for Frank-Wolfe-based methods that we are aware of. Moreover, while to the best of our knowledge, so called *away-steps* in Frank-Wolfe algorithms were only shown to facilitate linear convergence rates when optimizing over polytopes [19], here we establish that also for the feasible set of positive semidefinite matrices in Problem (2), such away-steps can lead to linear convergence rates (under proper assumptions), as we prove for our AFW and GAFW variants. An additional contribution is our connection between Frank-Wolfe methods and optimization over manifolds as captured by our GAFW variant which applies Frank-Wolfe steps and away-steps w.r.t. to the *geodesic gradient* of Problem (2). It is thus our hope that our work will lead to additional efficient Frank-Wolfe-based algorithms for other important and well-structured non-convex and potentially nonsmooth problems.

We introduce the following notation. We let $\mathbb{S}_+^p$ and $\mathbb{S}_{++}^p$ denote the sets of positive semidefinite and positive definite matrices in $\mathbb{S}^p$, respectively. We denote $\mathcal{S}_p = \{\mathbf{Q} \in \mathbb{S}_+^p \mid \mathrm{Tr}(\mathbf{Q}) = p\}$, $\mathcal{S}_{p+} = \{\mathbf{Q} \in \mathbb{S}_{++}^p \mid \mathrm{Tr}(\mathbf{Q}) = p\}$. For a matrix $\mathbf{M} \in \mathbb{S}^p$, we let $\lambda_i(\mathbf{M})$ denote the $i$th largest signed eigenvalue of $\mathbf{M}$. We also denote at times by $\lambda_{\max}(\mathbf{M})$ and $\lambda_{\min}(\mathbf{M})$ the largest and smallest (signed) eigenvalues, respectively. For real matrices we let $\|\cdot\|_2, \|\cdot\|_F, \|\cdot\|_1$ denote the spectral norm (largest singular value), Frobenius (Euclidean) norm, and trace norm (sum of singular values), respectively. We also denote by $\langle \cdot, \cdot \rangle$ the standard inner-product in $\mathbb{S}^p$.

It is known that for every $\mathbf{Q} \in \mathcal{S}_{p+}$, $f(\mathbf{Q})$ is finite, while for every sequence $(\mathbf{Q}_k)_{k \geq 1} \subseteq \mathcal{S}_{p+}$ with $\lambda_{\min}(\mathbf{Q}_k) \underset{k \to \infty}{\longrightarrow} 0$, it holds that $f(\mathbf{Q}_k) \underset{k \to \infty}{\longrightarrow} \infty$, see [28]. This implies the following lemma that will be used throughout the paper.

**Lemma 1.** *Fix $\mathbf{Q}_0 \in \mathcal{S}_{p+}$. There exists a constant $\lambda > 0$ such that for every $\mathbf{Q} \in \mathcal{S}_{p+}$ satisfying $f(\mathbf{Q}) \leq f(\mathbf{Q}_0)$, it holds that $\lambda_{\min}(\mathbf{Q}) \geq \lambda$.*

Table 1: Summary of main results. The bounds are given in simplified form excluding constants and logarithmic factors. We denote by $\kappa_0, \lambda_0, \tilde{\kappa}_0$ the maximal values of $\frac{\lambda_{\max}(\mathbf{Q})}{\lambda_{\min}(\mathbf{Q})}, \lambda_{\max}(\mathbf{Q}), \frac{\lambda_{\max}^2(\mathbf{Q})}{\lambda_{\min}(\mathbf{Q})}$ over the level set $\{\mathbf{Q} \in \mathcal{S}_p \mid f(\mathbf{Q}) \leq f(\mathbf{Q}_0)\}$, where $\mathbf{Q}_0$ is the initialization point, respectively. The $\nabla f$ notation denotes the gradient at the current point. $\rho$ is the constant of linear convergence (see Theorem 4). The linear rates are w.r.t. the approximation error in function value. The sublinear rates of AFW and GAFW are w.r.t. the distance in spectral norm from satisfying Eq. (1), while the sublinear rate for FW is for a related, yet slightly different measure, see Theorem 3.

| FW variant | single iteration runtime (Theorem 2) | sublinear rate (Theorem 3) | linear rate (Theorem 4) |
|---|---|---|---|
| FW | $(p^2 + np)\sqrt{\|\nabla f\|_2 \|\lambda_{\min}(\nabla f)\|^{-1}}$ | $\tilde{\kappa}_0^2/\epsilon^2$ | - |
| AFW | $p^2 + np$ | $\tilde{\kappa}_0^2/\epsilon^2$ | $\kappa_0^2 \rho^{-1} \log(1/\epsilon)$ |
| GAFW | $p^3 + np$ | $\lambda_0^2/\epsilon^2$ | $\rho^{-1} \log(1/\epsilon)$ |

## 2 Frank-Wolfe-based Algorithms for Approximating Tyler's M-estimator

We consider three Frank-Wolfe variants for Problem (2), all described below in Algorithm 1. All three variants require to solve (approximately) a certain eigenvalue problem, a different one for each variant, see more details below. All three variants have the following two properties: 1. they all use the same adaptive scheme to compute the step-size $\mu_t$, which in particular does not require any knowledge of the parameters of the problem (this choice will become clearer in the convergence analysis), and 2. since the gradient of the objective $f(\cdot)$ requires the inverse matrix $\mathbf{Q}_t^{-1}$, where $\mathbf{Q}_t$ is the current iterate of the algorithm, they all explicitly and efficiently maintain the inverse $\mathbf{Q}_t^{-1}$ using the well-known Sherman-Morrison formula and capitalizing on the fact that the algorithm performs a rank-one update. Hence, the inverse could be updated in only $O(p^2)$ time per iteration (as opposed to the standard $O(p^3)$ matrix inversion time). Below we expand on each of the eigenvalue problems in Algorithm 1 and how they correspond to familiar and new Frank-Wolfe-type methods.

We note that Algorithm 1 is completely independent of any parameter of Problem (2). The only parameter the algorithm accepts is the approximation parameter $\beta \in [0, 1)$, however this choice is not very important as will be evident from our convergence theorems, and only brought for convinience. In particular one can always simply choose some universal constant, for instance $\beta = 1/2$. The algorithm also requires a feasible initialization point $\mathbf{Q}_0 \in \mathcal{S}_{p+}$ and its inverse, and a convenient choice is to simply take $\mathbf{Q}_0 = \mathbf{I} = \mathbf{Q}_0^{-1}$.

---

**Algorithm 1** Frank-Wolfe variants for approximating Tyler's M-estimator

---

Input: $\mathbf{Q}_0, \mathbf{Q}_0^{-1}$ such that $\mathbf{Q}_0 \succ 0$, $\beta \in [0, 1)$
**for** t = 0, , ... **do**
    Solve an approximate eigenvalue problem: let $\mathbf{v}_t \in \mathbb{R}^p$ be such that $\|\mathbf{v}_t\| = \sqrt{p}$ and satisfies one of the following:

$$1. \quad -\mathbf{v}_t^\top \nabla f(\mathbf{Q}_t)\mathbf{v}_t \geq -p(1 - \beta)\lambda_{\min}(\nabla f(\mathbf{Q}_t)) \qquad \text{(FW step)} \qquad (5)$$

$$2. \quad |\mathbf{v}_t^\top \nabla f(\mathbf{Q}_t)\mathbf{v}_t| \geq p(1 - \beta)\|\nabla f(\mathbf{Q}_t)\|_2 \qquad \text{(AFW step)} \qquad (6)$$

$$3. \quad \frac{|\mathbf{v}_t^\top \nabla f(\mathbf{Q}_t)\mathbf{v}_t|}{\mathbf{v}_t^\top \mathbf{Q}_t^{-1}\mathbf{v}_t} \geq (1 - \beta)\|\mathbf{Q}_t^{1/2}\nabla f(\mathbf{Q}_t)\mathbf{Q}_t^{1/2}\|_2 \qquad \text{(GAFW step)} \qquad (7)$$

$\mu_t \leftarrow \frac{-\mathbf{v}_t^\top \nabla f(\mathbf{Q}_t)\mathbf{v}_t}{(\mathbf{v}_t^\top \mathbf{Q}_t^{-1}\mathbf{v}_t)^2 - \mathbf{v}_t^\top \nabla f(\mathbf{Q}_t)\mathbf{v}_t}$
$\gamma_t \leftarrow \frac{\mu_t}{1-\mu_t} = \frac{-\mathbf{v}_t^\top \nabla f(\mathbf{Q}_t)\mathbf{v}_t}{(\mathbf{v}_t^\top \mathbf{Q}_t^{-1}\mathbf{v}_t)^2}$
$\mathbf{Q}_{t+1} \leftarrow \mathbf{Q}_t + \mu_t(\mathbf{v}_t\mathbf{v}_t^\top - \mathbf{Q}_t)$
$\mathbf{Q}_{t+1}^{-1} \leftarrow \frac{1}{1-\mu_t}(\mathbf{Q}_t^{-1} - \gamma_t \frac{\mathbf{Q}_t^{-1}\mathbf{v}_t\mathbf{v}_t^\top \mathbf{Q}_t^{-1}}{1+\gamma_t \mathbf{v}_t^\top \mathbf{Q}_t^{-1}\mathbf{v}_t})$
**end for**

---

**Standard Frank-Wolfe update:** For the feasible set $\mathcal{S}_p$, it is well known that the solution to the FW linear optimization problem $\min_{\mathbf{V} \in \mathcal{S}_p}\langle \mathbf{V}, \nabla f(\mathbf{Q}_t)\rangle$ at the current point $\mathbf{Q}_t$ is given w.l.o.g. as $\mathbf{V}_+ := \mathbf{v}\mathbf{v}^\top$, where $\mathbf{v} = \sqrt{p}\mathbf{u}$ and $\mathbf{u}$ is a unit-length eigenvector of $\nabla f(\mathbf{Q}_t)$ corresponding to the

smallest eigenvalue, i.e., $\mathbf{u}^\top \nabla f(\mathbf{Q})\mathbf{u} = \lambda_{\min}(\nabla f(\mathbf{Q}_t))$, see for instance [14, 15]. This is exactly the step in Eq. (5) in Algorithm 1, only that in Eq. (5) we do not require exact computation of the eigenvector, but allow for a $(1 - \beta)$ multiplicative approximation w.r.t. the smallest eigenvalue.

**Frank-Wolfe with Away-Steps (AFW) update:** The AFW update (see [13, 19]) applies one of two types of updates. One is the standard FW step discussed above. The other, called *away-step*, takes the form $\mathbf{Q}_{t+1} \leftarrow \mathbf{Q}_t + \mu(\mathbf{Q}_t - \mathbf{V}_-)$, $\mu > 0$, where $\mathbf{V}_- \in \mathcal{S}_p$ is ideally chosen so that $\mathbf{V}_-$ maximizes the inner product $\langle \mathbf{V}_-, \nabla f(\mathbf{Q}_t) \rangle$ over $\mathcal{S}_p$, subject to the constraint that there indeed exists a corresponding $\mu > 0$, so that $\mathbf{Q}_{t+1}$ is feasible [13, 19]. For $\mathbf{Q}_t \succ 0$, all points in $\mathcal{S}_p$ give rise to such positive $\mu$, and so, $\mathbf{V}_-$ is simply the solution to $\max_{\mathbf{V} \in \mathcal{S}_p} \langle \mathbf{V}, \nabla f(\mathbf{Q}_t) \rangle$, which w.l.o.g. is of the form $\mathbf{V}_- = \mathbf{v}\mathbf{v}^\top$, where $\mathbf{v} = \sqrt{p}\mathbf{u}$ and $\mathbf{u}$ is a unit-length eigenvector of $\nabla f(\mathbf{Q}_t)$ corresponding to the largest (signed) eigenvalue.

If $\langle \mathbf{Q}_t - \mathbf{V}_+, \nabla f(\mathbf{Q}_t) \rangle \geq \langle \mathbf{V}_- - \mathbf{Q}_t, \nabla f(\mathbf{Q}_t) \rangle$, AFW performs a standard FW update with $\mathbf{V}_+$, and otherwise, it performs an away-step with $\mathbf{V}_-$. In case of Problem (2), a straightforward calculation shows that $\langle \mathbf{Q}_t, \nabla f(\mathbf{Q}_t) \rangle = 0$. Thus, in our case, a FW step is taken if $-\langle \mathbf{V}_+, \nabla f(\mathbf{Q}_t) \rangle \geq \langle \mathbf{V}_-, \nabla f(\mathbf{Q}_t) \rangle$, i.e., if $-p\lambda_{\min}(\nabla f(\mathbf{Q}_t)) \geq p\lambda_{\max}(\nabla f(\mathbf{Q}_t))$, and otherwise an away-step is taken. Thus, taking both cases into account, AFW performs a step of the form $\mathbf{Q}_{t+1} \leftarrow \mathbf{Q}_t + \mu(\mathbf{v}\mathbf{v}^\top - \mathbf{Q}_t)$ (here $\mu$ can also be negative), where $|\mathbf{v}^\top \nabla f(\mathbf{Q}_t)\mathbf{v}| \geq p\max\{-\lambda_{\min}(\nabla f(\mathbf{Q}_t)), \lambda_{\max}(\nabla f(\mathbf{Q}_t))\} = p\|\nabla f(\mathbf{Q}_t)\|_2$, which is exactly what we have in Eq. (6) in Algorithm 1, only that in (6) we again do not require precise computation, but allow for a $(1 - \beta)$ multiplicative approximation.

**Geodesic Frank-Wolfe with Away-Steps (GAFW) update:** GAFW performs the same updates as AFW, but not w.r.t. to the (standard) gradient $\nabla f(\mathbf{Q}_t)$, but with respect to the so-called *geodesic gradient* given by $\mathbf{Q}_t^{1/2} \nabla f(\mathbf{Q}_t) \mathbf{Q}_t^{1/2}$ (see for instance [6]). Note that $\|\mathbf{Q}_t^{1/2} \nabla f(\mathbf{Q}_t) \mathbf{Q}_t^{1/2}\|_2 = \max_{\mathbf{u}:\|\mathbf{u}\|_2 = 1} |\mathbf{u}^\top \mathbf{Q}_t^{1/2} \nabla f(\mathbf{Q}_t) \mathbf{Q}_t^{1/2} \mathbf{u}|_2$. Consider the parametrization $\mathbf{u} = \frac{\mathbf{Q}_t^{-1/2}\mathbf{v}}{\|\mathbf{Q}_t^{-1/2}\mathbf{v}\|_2}$ (note it is invariant to the norm of $\mathbf{v}$). This gives $\|\mathbf{Q}_t^{1/2} \nabla f(\mathbf{Q}_t) \mathbf{Q}_t^{1/2}\|_2 = \max_{\mathbf{v}} \frac{|\mathbf{v}^\top \nabla f(\mathbf{Q}_t)\mathbf{v}|}{\mathbf{v}^\top \mathbf{Q}_t^{-1}\mathbf{v}}$, which is exactly the update step in Eq. (7) in Algorithm 1, only that in Eq. (7) it suffices to find a $(1 - \beta)$-multiplicative approximation. As will be evident from our analysis, and in particular in the error reduction argument in Lemma 3, this step maximizes the decrease in function value on each iteration.

The following lemma establishes that the step-sizes of Algorithm 1 indeed always produce feasible iterates in $\mathcal{S}_{p+}$. The proof is given in the appendix.

**Lemma 2** (Feasibility of Algorithm 1). *Suppose that $Tr(\mathbf{Q}_0) = p$ and $\mathbf{Q}_0 \succ 0$. Then, for every iteration $t \geq 1$ of Algorithm 1 it holds that $Tr(\mathbf{Q}_t) = p$ and $\mathbf{Q}_t \succ 0$.*

### 2.1 Efficient implementation of the eigenvalue oracles in Algorithm 1

We now discuss how the various eigenvalue problems in Algorithm 1 could be solved (to sufficient approximation) efficiently via simple and well-known iterative algorithms for leading eigenvector computation, such as the *power method* or the *Lanczos algorithm* [12, 23]. These implementations rely on three simple ideas:

1. Use of the Sherman-Morrison formula for rank-one updates to update the inverse matrix $\mathbf{Q}_{t+1}^{-1}$ from the previous one $\mathbf{Q}_t^{-1}$, given the vector $\mathbf{v}_t$, in $O(p^2)$ time.

2. Explicitly maintaining and updating the vectors $\mathbf{Q}_t^{-1}\mathbf{x}_i, i = 1, \ldots, n$, which can be done in overall $O(np)$ time per iteration. This allows to compute a matrix-vector product of the form $\nabla f(\mathbf{Q}_t)\mathbf{v}$, for some $\mathbf{v} \in \mathbb{R}^p$, in only $O(np)$ time (see expression for $\nabla f$ in Eq. (3)).

3. Apply the power method or the Lanczos algorithm (or any other method for leading eigenvector computation) to solve the leading eigenvalue problems (5), (6), (7), relying on the fact that due to the pervious two items, each iteration of the power method requires only $O(p^2 + np)$ time.

The complete proof of the following theorem is given in the appendix.

**Theorem 2.** *Let $\beta \in [0, 1)$ be some universal constant (e.g., $\beta = 1/2$). Algorithm 1 admits implementations based on fast algorithms for leading eigenvector computation (e.g., the power*

*method / Lanczos [12, 23]), such that each iteration of Algorithm 1 could be implemented in:
$\tilde{O}(np + p^2)$ time when using AFW steps (Eq. (6)), $O(p^3) + \tilde{O}(np + p^2)$ time when using GAFW
steps (Eq. (7)), and $\tilde{O}\left(\sqrt{\frac{\|\nabla f(\mathbf{Q}_t)\|_2}{|\lambda_p(\nabla f(\mathbf{Q}_t))|}}(np + p^2)\right)$ when using FW steps (Eq. (5)), where in all cases
the $\tilde{O}$ notation hides a logarithmic factor in the dimension $p$ and the probability of failure $\delta$.*

**Remark 1.** *Note that the worst-case time to solve the standard FW eigenvalue problem, in terms
of the dependence on the size of the data $np$, is much worse than that of AFW and GAFW. This is
because, while the eigenvalue problems in AFW, GAFW require to approximate the largest eigenvalue
in magnitude, FW requires to approximate the smallest signed eigenvalue.*

**Remark 2.** *Since leading eigenvector algorithms such as the power method are usually initialized
with a random vector, their guarantees only hold with high probability, as captured in Theorem
2. However, since the dependence on the probability of failure is only logarithmic, for the clarity
of presentation, henceforth we neglect such considerations and treat these computations as if they
always succeed.*

# 3 Sublinear Convergence of Algorithm 1

While the measure of convergence in our sublinear rates for AFW and GAFW will be simply the
distance in spectral norm from satisfying the TME equation (1) , our measure of convergence for the
standard FW variant is slightly less obvious and is motivated by the following observation, the proof
of which is given in the appendix.

**Observation 1.** *For any* $\mathbf{Q} \in \mathcal{S}_{p+}$, $\lambda_{\min}\left(\mathbf{Q} - \frac{p}{n}\sum_{i=1}^n \frac{\mathbf{x}_i\mathbf{x}_i^\top}{\mathbf{x}_i^\top \mathbf{Q}^{-1}\mathbf{x}_i}\right) \leq 0$, *and*
$\lambda_{\min}\left(\mathbf{Q} - \frac{p}{n}\sum_{i=1}^n \frac{\mathbf{x}_i\mathbf{x}_i^\top}{\mathbf{x}_i^\top \mathbf{Q}^{-1}\mathbf{x}_i}\right) = 0$ *if and only if* $\mathbf{Q} = \mathbf{Q}^*$.

**Theorem 3** ($O(1/\epsilon^2)$ convergence of Algorithm 1)**.** *Consider the iterates of Algorithm 1 and define
the function $T(\tilde{\epsilon}) = \lceil 4(f(\mathbf{Q}_0) - f(\mathbf{Q}^*))(1 + \tilde{\epsilon}^{-2})\rceil$. Fix $\epsilon > 0$ and define*

$$\tilde{\epsilon}_{FW} := \left(\min_{\mathbf{Q}\in\mathcal{S}_p:f(\mathbf{Q})\leq f(\mathbf{Q}_0)} \frac{\lambda_{\min}(\mathbf{Q})}{\lambda_{\max}^2(\mathbf{Q})}\right)(1-\beta)\epsilon, \; \tilde{\epsilon}_{AFW} := \left(\min_{\mathbf{Q}\in\mathcal{S}_p:f(\mathbf{Q})\leq f(\mathbf{Q}_0)} \frac{\lambda_{\min}(\mathbf{Q})}{\lambda_{\max}^2(\mathbf{Q})}\right)(1-\beta)\epsilon,$$

$$\tilde{\epsilon}_{GAFW} := \left(\max_{\mathbf{Q}\in\mathcal{S}_p:f(\mathbf{Q})\leq f(\mathbf{Q}_0)} \lambda_{\max}(\mathbf{Q})\right)^{-1}(1-\beta)\epsilon.$$

*Then, when using FW steps (Eq. 5), it holds that for all $t \geq T(\tilde{\epsilon}_{FW})$,*

$$\max_{\tau=0,\dots,t-1} \lambda_{\min}\left(\mathbf{Q}_\tau - \frac{p}{n}\sum_{i=1}^n \frac{\mathbf{x}_i\mathbf{x}_i^\top}{\mathbf{x}_i^\top \mathbf{Q}_\tau^{-1}\mathbf{x}_i}\right) \geq -\epsilon. \tag{8}$$

*When using AFW steps (Eq. (6)) or GAFW steps (Eq. (7)), it holds for all $t \geq T(\tilde{\epsilon}_{AFW})$ or
$t \geq T(\tilde{\epsilon}_{GAFW})$, respectively, that*

$$\min_{\tau=0,\dots,t-1} \left\|\mathbf{Q}_\tau - \frac{p}{n}\sum_{i=1}^n \frac{\mathbf{x}_i\mathbf{x}_i^\top}{\mathbf{x}_i^\top \mathbf{Q}_\tau^{-1}\mathbf{x}_i}\right\|_2 \leq \epsilon. \tag{9}$$

**Remark 3.** *Note that one motivation for considering the use of the GAFW variant in light of
Theorem 3, is that it enjoys better conditioning than AFW in terms of the maximal condition number
$\lambda_{\max}(\mathbf{Q})/\lambda_{\min}(\mathbf{Q})$ over the initial level set $\{\mathbf{Q} \in \mathcal{S}_p \mid f(\mathbf{Q}) \leq f(\mathbf{Q}_0)\}$.*

The complete proof of Theorem 3 is given in the appendix. Below we state and prove the key
technical step — a bound on the improvement in function value that Algorithm 1 makes on a single
iteration. In particular, the lemma is independent of the way the vector $\mathbf{v}_t$ is generated and hence
applies to all three variants described in Algorithm 1.

**Lemma 3.** *Fix some iteration $t$ of Algorithm 1 and define $L_t = \frac{\mathbf{v}_t^\top \nabla f(\mathbf{Q}_t)\mathbf{v}_t}{\mathbf{v}_t^\top \mathbf{Q}_t^{-1}\mathbf{v}_t}$. It holds that*

$$f(\mathbf{Q}_{t+1}) - f(\mathbf{Q}_t) \leq -\frac{1}{4}\min\{1, L_t^2\}. \tag{10}$$

**Remark 4.** *An immediate important consequence of Lemma 3 is that Algorithm 1 is a decent method, i.e., the function value never increases from one iteration to the next. Lemma 3 also reveals one motivation for the GAFW step in Eq. (7): (up to a $(1 - \beta)$ factor) it maximizes the reduction in function value.*

*Proof of Lemma 3.* Using the definition of $\mathbf{Q}_{t+1}$ we have that,

$$f(\mathbf{Q}_{t+1}) = \frac{p}{n} \sum_{i=1}^{n} \log(\mathbf{x}_i^\top ((1 - \mu_t)\mathbf{Q}_t + \mu_t \mathbf{v}_t \mathbf{v}_t^\top)^{-1} \mathbf{x}_i) + \log(\det((1 - \mu_t)\mathbf{Q}_t + \mu_t \mathbf{v}_t \mathbf{v}_t^\top)). \tag{11}$$

Using the Sherman-Morrison formula we have that,

$$((1 - \mu_t)\mathbf{Q}_t + \mu_t \mathbf{v}_t \mathbf{v}_t^\top)^{-1} = \frac{1}{1 - \mu_t} \left( \mathbf{Q}_t^{-1} - \gamma_t \frac{\mathbf{Q}_t^{-1} \mathbf{v}_t \mathbf{v}_t^\top \mathbf{Q}_t^{-1}}{1 + \gamma_t \mathbf{v}_t^\top \mathbf{Q}_t^{-1} \mathbf{v}_t} \right), \tag{12}$$

where we recall that $\gamma_t = \frac{\mu_t}{1 - \mu_t}$.

Using the well-known matrix determinant lemma for rank-one updates we have,

$$\det((1 - \mu_t)\mathbf{Q}_t + \mu_t \mathbf{v}_t \mathbf{v}_t^\top) = (1 - \mu_t)^p (1 + \gamma_t \mathbf{v}_t^\top \mathbf{Q}_t^{-1} \mathbf{v}_t) \det(\mathbf{Q}_t). \tag{13}$$

Plugging (12) and (13) into (11), we obtain

$$
\begin{aligned}
f(\mathbf{Q}_{t+1}) &= \frac{p}{n} \sum_{i=1}^{n} \log \left( \frac{1}{1 - \mu_t} \mathbf{x}_i^\top \left( \mathbf{Q}_t^{-1} - \gamma_t \frac{\mathbf{Q}_t^{-1} \mathbf{v}_t \mathbf{v}_t^\top \mathbf{Q}_t^{-1}}{1 + \gamma_t \mathbf{v}^\top \mathbf{Q}_t^{-1} \mathbf{v}_t} \right) \mathbf{x}_i \right) \\
&\quad + \log \left( (1 - \mu_t)^p (1 + \gamma_t \mathbf{v}_t^\top \mathbf{Q}_t^{-1} \mathbf{v}_t) \det(\mathbf{Q}_t) \right) \\
&= \frac{p}{n} \sum_{i=1}^{n} \log \left( \mathbf{x}_i^\top \left( \mathbf{Q}_t^{-1} - \gamma_t \frac{\mathbf{Q}_t^{-1} \mathbf{v}_t \mathbf{v}_t^\top \mathbf{Q}_t^{-1}}{1 + \gamma \mathbf{v}_t^\top \mathbf{Q}_t^{-1} \mathbf{v}_t} \right) \mathbf{x}_i \right) + \log \left( (1 + \gamma_t \mathbf{v}^\top \mathbf{Q}_t^{-1} \mathbf{v}_t) \det(\mathbf{Q}_t) \right) \\
&= \frac{p}{n} \sum_{i=1}^{n} \log \left( \mathbf{x}_i^\top \mathbf{Q}_t^{-1} \mathbf{x}_i - \gamma_t \frac{(\mathbf{x}_i^\top \mathbf{Q}_t^{-1} \mathbf{v}_t)^2}{1 + \gamma_t \mathbf{v}_t^\top \mathbf{Q}_t^{-1} \mathbf{v}_t} \right) + \log(1 + \gamma_t \mathbf{v}_t^\top \mathbf{Q}_t^{-1} \mathbf{v}_t) + \log \det(\mathbf{Q}_t) \\
&= \frac{p}{n} \sum_{i=1}^{n} \left( \log(\mathbf{x}_i^\top \mathbf{Q}_t^{-1} \mathbf{x}_i) + \log \left( 1 - \gamma_t \frac{1}{1 + \gamma_t \mathbf{v}_t^\top \mathbf{Q}_t^{-1} \mathbf{v}_t} \frac{(\mathbf{x}_i^\top \mathbf{Q}_t^{-1} \mathbf{v}_t)^2}{\mathbf{x}_i^\top \mathbf{Q}_t^{-1} \mathbf{x}_i} \right) \right) \\
&\quad + \log(1 + \gamma_t \mathbf{v}_t^\top \mathbf{Q}_t^{-1} \mathbf{v}_t) + \log \det(\mathbf{Q}_t) \\
&= f(\mathbf{Q}_t) + \frac{p}{n} \sum_{i=1}^{n} \log \left( 1 - \gamma_t \frac{1}{1 + \gamma_t \mathbf{v}_t^\top \mathbf{Q}_t^{-1} \mathbf{v}_t} \frac{(\mathbf{x}_i^\top \mathbf{Q}_t^{-1} \mathbf{v}_t)^2}{\mathbf{x}_i^\top \mathbf{Q}_t^{-1} \mathbf{x}_i} \right) + \log(1 + \gamma_t \mathbf{v}_t^\top \mathbf{Q}_t^{-1} \mathbf{v}_t) \\
&\underset{(a)}{\leq} f(\mathbf{Q}_t) - \frac{\gamma_t}{1 + \gamma_t \mathbf{v}_t^\top \mathbf{Q}_t^{-1} \mathbf{v}_t} \frac{p}{n} \sum_{i=1}^{n} \frac{(\mathbf{x}_i^\top \mathbf{Q}_t^{-1} \mathbf{v}_t)^2}{\mathbf{x}_i^\top \mathbf{Q}_t^{-1} \mathbf{x}_i} + \log(1 + \gamma_t \mathbf{v}_t^\top \mathbf{Q}_t^{-1} \mathbf{v}_t) \\
&= f(\mathbf{Q}_t) + \frac{\gamma_t}{1 + \gamma_t \mathbf{v}_t^\top \mathbf{Q}_t^{-1} \mathbf{v}_t} \left( \mathbf{v}_t^\top \nabla f(\mathbf{Q}_t) \mathbf{v}_t - \mathbf{v}_t^\top \mathbf{Q}_t^{-1} \mathbf{v}_t \right) + \log(1 + \gamma_t \mathbf{v}_t^\top \mathbf{Q}_t^{-1} \mathbf{v}_t),
\end{aligned}
$$

where (a) follows from the inequality $\log(1 + x) \leq x$.

We now consider two cases. If $\gamma_t \geq 0$, using the inequality $\log(1 + x) \leq \frac{x}{2} \frac{2+x}{1+x} = x - \frac{x^2}{2(1+x)}$ for all $x \geq 0$, we have that

$$\log(1 + \gamma_t \mathbf{v}_t^\top \mathbf{Q}_t^{-1} \mathbf{v}_t) \leq \gamma_t \mathbf{v}_t^\top \mathbf{Q}_t^{-1} \mathbf{v}_t - \frac{\gamma_t^2 (\mathbf{v}_t^\top \mathbf{Q}_t^{-1} \mathbf{v}_t)^2}{2(1 + \gamma_t \mathbf{v}_t^\top \mathbf{Q}_t^{-1} \mathbf{v}_t)}.$$

If $\gamma_t < 0$, using the inequality $\log(1 + x) \leq \frac{2x}{2+x} = x - \frac{x^2}{2+x}$ for all $0 \geq x > -1$, we have that

$$\log(1 + \gamma_t \mathbf{v}_t^\top \mathbf{Q}_t^{-1} \mathbf{v}_t) \leq \gamma_t \mathbf{v}_t^\top \mathbf{Q}_t^{-1} \mathbf{v}_t - \frac{\gamma_t^2 (\mathbf{v}_t^\top \mathbf{Q}_t^{-1} \mathbf{v}_t)^2}{2 + \gamma_t \mathbf{v}_t^\top \mathbf{Q}_t^{-1} \mathbf{v}_t}.$$

It is easily verified that for our choice $\gamma_t = \frac{-\mathbf{v}_t^\top \nabla f(\mathbf{Q}_t)\mathbf{v}_t}{(\mathbf{v}_t^\top \mathbf{Q}_t^{-1}\mathbf{v}_t)^2}$, it holds that

$$\frac{\gamma_t}{1 + \gamma_t \mathbf{v}_t^\top \mathbf{Q}_t^{-1}\mathbf{v}_t} \left(\mathbf{v}_t^\top \nabla f(\mathbf{Q}_t)\mathbf{v}_t - \mathbf{v}_t^\top \mathbf{Q}_t^{-1}\mathbf{v}_t\right) + \gamma_t \mathbf{v}_t^\top \mathbf{Q}_t^{-1}\mathbf{v}_t = 0.$$

Thus, considering both options for $\gamma_t$ ($\geq 0$ or $< 0$), we have that

$$f(\mathbf{Q}_{t+1}) - f(\mathbf{Q}_t) \leq -\frac{\gamma_t^2 (\mathbf{v}_t^\top \mathbf{Q}_t^{-1}\mathbf{v}_t)^2}{2(1 + |\gamma_t \mathbf{v}_t^\top \mathbf{Q}_t^{-1}\mathbf{v}_t|)} = -\frac{L_t^2}{2(1 + |L_t|)}.$$

The Lemma follows from considering the two cases $|L_t| \geq 1$ and $|L_t| < 1$, and simplifying.

$\square$

**Corollary 1.** *Fix $\epsilon > 0$. For any $t \geq \left\lceil 4(f(\mathbf{Q}_0) - f(\mathbf{Q}^*))\left(1 + \epsilon^{-2}\right)\right\rceil$ iterations of Algorithm 1 it holds that,* $\min_{\tau=0,\ldots,t-1} |L_\tau| \leq \epsilon$.

*Proof.* Fix some iteration $t$ of Algorithm 1. Using Lemma 3 we have that

$$f(\mathbf{Q}^*) - f(\mathbf{Q}_0) \leq f(\mathbf{Q}_t) - f(\mathbf{Q}_0) = \sum_{\tau=0}^{t-1} f(\mathbf{Q}_{\tau+1}) - f(\mathbf{Q}_\tau) \leq -\frac{1}{4}\sum_{\tau=0}^{t-1} \min\{1, L_\tau^2\}.$$

Thus, for $t \geq \left\lceil 4(f(\mathbf{Q}_0) - f(\mathbf{Q}^*))\left(1 + \epsilon^{-2}\right)\right\rceil$ iterations there must exist some $\tau \in \{0,\ldots,t-1\}$ such that $|L_\tau| \leq \epsilon$. $\square$

## 4 Linear Convergence of AFW and GAFW

In this section we prove that under an additional (to Assumption 1) mild assumption, the AFW and GAFW variants converge linearly in function value.

**Assumption 2.** *The data-points $\mathbf{x}_1, \ldots, \mathbf{x}_n$ satisfy that $n \geq 2p$, and for any subset $\mathcal{S} \subseteq \{\mathbf{x}_1, \ldots, \mathbf{x}_n\}$ such that $|\mathcal{S}| \geq n/2$, it holds that $span(\mathcal{S}) = \mathbb{R}^p$.*

**Remark 5.** *Note that for any fixed $n$, if the (normalized to have unit norms) data-points are i.i.d. samples from a continuous distribution which is supported on the entire unit sphere, Assumption 2 holds with probability 1.*

**Theorem 4** (Linear convergence of AFW and GAFW). *Denote $h_t = f(\mathbf{Q}_t) - f(\mathbf{Q}^*)$ for all $t \geq 0$. Suppose Assumption 2 holds. Then, Algorithm 1 when run with AFW steps, satisfies*

$$\forall t \geq 0 : \quad h_t \leq h_0 \exp\left(-\frac{(1-\beta)^2}{4}\kappa_0^{-2}\rho\left(t - \lceil 4(f(\mathbf{Q}_0) - f(\mathbf{Q}^*))\rceil\right)\right),$$

*where $\rho > 0$ is the constant implied by Theorem 5, and $\kappa_0 := \max_{\mathbf{Q}\in\mathcal{S}_p : f(\mathbf{Q})\leq f(\mathbf{Q}_0)} \frac{\lambda_{\max}(\mathbf{Q})}{\lambda_{\min}(\mathbf{Q})}$.*

*When using GAFW steps, Algorithm 1 satisfies*

$$\forall t \geq 0 : \quad h_t \leq h_0 \exp\left(-\frac{(1-\beta)^2}{4}\rho\left(t - \lceil 4(f(\mathbf{Q}_0) - f(\mathbf{Q}^*))\rceil\right)\right).$$

**Remark 6.** *As with Theorem 3, we see that GAFW enjoys better conditioning than AFW w.r.t. the maximal condition number $\lambda_{\max}(\mathbf{Q})/\lambda_{\min}(\mathbf{Q})$ over the initial level set.*

The complete proof of Theorem 4 is given in the appendix. The key part in the proof is to establish that, under Assumption 2, Problem (2) satisfies a Polyak-Łojasiewicz (PL) condition w.r.t. both the standard gradient $\nabla f(\mathbf{Q})$ and the geodesic gradient $\mathbf{Q}^{1/2}\nabla f(\mathbf{Q})\mathbf{Q}^{1/2}$, which is a property well-known to facilitate linear convergence rates for first-order methods, see for instance [22, 17, 9]. A (standard, i.e., non geodesic) PL condition at some query point $\mathbf{Q}$ takes the form $\|\nabla f(\mathbf{Q})\|^2 \geq C(f(\mathbf{Q}) - f(\mathbf{Q}^*))$, for some constant $C > 0$ independent of $\mathbf{Q}$.

The proof of our PL condition is highly non-trivial and is very much inspired by the analysis in [6]. However, while the proof in [6] relies heavily on *geodesic strong convexity* and *quantum expansion* arguments, we give a different proof (in particular, our Assumption 2 is different from their explicit assumption that the data is sampled from an elliptical distribution) that does not use such considerations and, we believe, is more straightforward and accessible.

**Theorem 5** (Polyak-Łojasiewicz condition). *Suppose Assumption 2 holds, and let $\mathbf{Q}_0 \in \mathcal{S}_{p+}$. Then, there exists a constant $\rho > 0$ such that for any $\mathbf{Q} \in \mathcal{S}_{p+}$ satisfying $f(\mathbf{Q}) \leq f(\mathbf{Q}_0)$, it holds that,*

$$\|\mathbf{Q}^{1/2}\nabla f(\mathbf{Q})\mathbf{Q}^{1/2}\|_2^2 \geq \rho\left(f(\mathbf{Q}) - f(\mathbf{Q}^*)\right).$$

**Remark 7.** *While Theorem 5 establishes that the PL parameter $\rho$ is strictly positive, its precise value (i.e., its dependence on the data) is quite intricate and does not admit a simple formula (see proof in appendix for more details).*

## 5 Numerical Simulations

In order to give some demonstration for the empirical performance of our Frank-Wolfe-based algorithms, we conducted two types of experiments that closely follow those in [28] (Chapter 3) with some minor changes, which consider a Gaussian distribution with outlier contamination, and a heavy-tailed multivariate t-distribution. We consider a large sample regime in which $n = p^2$. We recall that in this regime each iteration of the Fixed-point Iterations method (FPI) takes $O(np^2)$ time, while our AFW, GAFW variants require only $O(np)$ time, up to a single log term, per-iteration.

**Data generation:** Following the experiments conducted in [28] with some minor changes, we consider i. a Gaussian distribution with outliers contamination in which, each Gaussian-distributed vector is replaced with probability $0.9/p$ with the eigenvector associated with the smallest eigenvalue of the covariance, and ii. a heavy-tailed multivariate t-distribution with two degrees of freedom. In both experiments we set $p = 50, n = 2500$, and we take the true unknown covariance to be a Toeplitz matrix with the elements $\mathbf{Q}_{i,j} = 0.85^{|i-j|}$.

**Methodology:** To present results that are implementation and scale independent as possible, for each of the considered methods we estimate the required running time per number of iterations, normalized by the data-size $np$. Since we consider the regime $n = p^2$, each iteration of FPI takes $O(np^2)$ time. For all Frank-Wolfe variants (FW, AFW, and GAFW) we use Python's SCIPY.SPARSE.LINALG.EIGSH procedure, which is based on the Lanczos algorithm, to solve the corresponding eigenvalue problem to low-accuracy (which corresponds to the approximation parameter $\beta$ in Algorithm 1). We verified that in all of our experiments and for all FW variants, this procedure makes at most 2 iterations, and thus, per the discussion in Section 2.1, the runtime per iteration of each of these variants is estimated by $O(np)$. Thus, normalizing the estimated runtime by the data-size $np$, in the figures below each iteration of FPI is estimated to take $p$ times more than that of any of the FW variants. All methods are initialized from the sample covariance (normalized to have trace equals $p$) and each figure is the average of 20 i.i.d. experiments.

**Results:** For all experiments we compute Tyler's estimator to high accuracy by running 250 iterations of the Fixed-point method and we denote the resulting matrix by $\mathbf{Q}^*$. In Figure 1 we report the distance in spectral norm (in log scale) of the iterates of the different methods form $\mathbf{Q}^*$, and in Figure 2 we report the approximation error $f(\mathbf{Q}_t) - f(\mathbf{Q}^*)$ of the iterates (also in log scale). It can be seen that in both setups and with respect to both measures, GAFW converges faster than FPI, and that in the contaminated Gaussian distribution setup it is significantly faster. Moreover, looking at the approximation error in log-scale, it indeed seems to exhibit a linear convergence rate. We also observe that the FW and AFW variants converge significantly slower than GAFW, which demonstrates how the better conditioning of GAFW (as captured also in our convergence theorems, Theorems 3 and 4) may be significant. Additionally, and perhaps surprisingly, AFW converges slower than FW, which suggests a situation in which AFW takes many *away-steps*, since they are better descent directions than the standard FW step, but in turn results in much smaller step-sizes, and thus overall, the convergence is slower.

## 6 Conclusions

We have presented the first Frank-Wolfe-based variants for approximating Tyler's M-estimator for robust and heavy-tailed covariance estimation. In particular these include parameter-free and globally-convergent variants with nearly linear runtime per-iteration and, under a mild assumption, with linear convergence rates, despite the fact that the underlying optimization problem is not convex

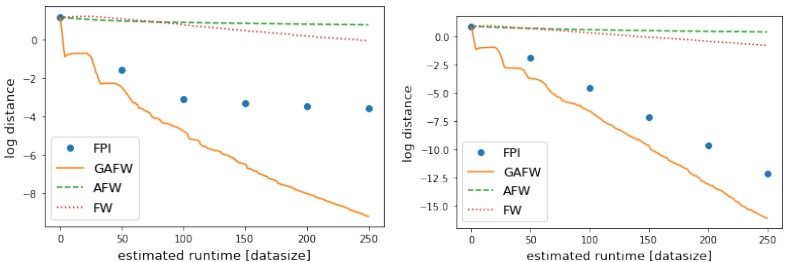

Figure 1: $\log \|\mathbf{Q}_t - \mathbf{Q}^*\|_2$ for Gaussian distribution with outlier contamination (left panel) and for heavy-tailed t-distribution (right panel).

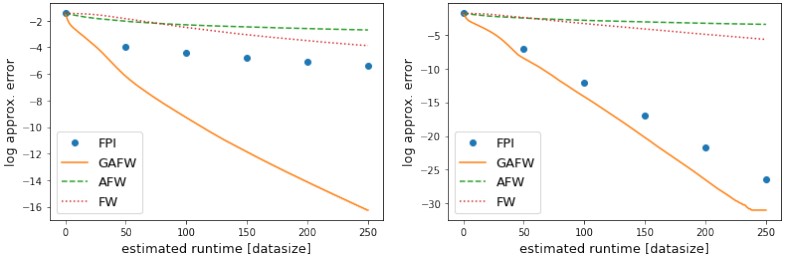

Figure 2: Log approximation error $f(\mathbf{Q}_t) - f(\mathbf{Q}^*)$ for Gaussian distribution with outlier contamination (left panel) and for heavy-tailed t-distribution (right panel).

nor smooth. We hope these results will pave the way to nearly linear-time algorithms for additional highly-structured nonconvex and nonsmooth problems.

## Acknowledgements

This research was supported by the ISRAEL SCIENCE FOUNDATION (grant No. 2267/22).

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
