# A  Proofs Missing from Section 2

## A.1  Proof of Lemma 2

Before proving the lemma we need a simple observation.

**Observation 2.** *For any $\mathbf{Q} \in \mathcal{S}_{p+}$ and any vector $\mathbf{v} \in \mathbb{R}^p$ such that $\|\mathbf{v}\| = \sqrt{p}$, we have that*

$$(\mathbf{v}^\top \mathbf{Q}^{-1} \mathbf{v})^2 - \mathbf{v}^\top \nabla f(\mathbf{Q})\mathbf{v} > 0.$$

*Proof.* Since $\mathbf{Q} \succ 0$, using the expression for $\nabla f(\mathbf{Q})$ in Eq. (3), we have that

$$(\mathbf{v}^\top \mathbf{Q}^{-1} \mathbf{v})^2 - \mathbf{v}^\top \nabla f(\mathbf{Q})\mathbf{v} \geq (\mathbf{v}^\top \mathbf{Q}^{-1} \mathbf{v})^2 - \mathbf{v}^\top \mathbf{Q}^{-1}\mathbf{v}.$$

Thus, for the condition stated in the observation to hold true, it suffices that $\mathbf{v}^\top \mathbf{Q}^{-1}\mathbf{v} > 1$. Since $\|\mathbf{v}\| = \sqrt{p}$ we have that

$$\mathbf{v}^\top \mathbf{Q}^{-1}\mathbf{v} \geq p\lambda_{\min}(\mathbf{Q}^{-1}) = \frac{p}{\lambda_{\max}(\mathbf{Q})} > \frac{p}{p},$$

where the last inequality follows since $\mathrm{Tr}(\mathbf{Q}) = p$ and $\mathbf{Q} \succ 0$, which imply that $\lambda_{\max}(\mathbf{Q}) < p$.

Thus, the observation is indeed correct. $\qquad\qquad\square$

*Proof of Lemma 2.* The fact that for all $t$ it holds that $\mathrm{Tr}(\mathbf{Q}_t) = p$ follows immediately from the definition of $\mathbf{Q}_{t+1}$ in Algorithm 1 and the fact that $\mathrm{Tr}(\mathbf{v}_t\mathbf{v}_t^\top) = p$.

In order to show that for all $t$, $\mathbf{Q}_t \succ 0$, we consider two cases. First, suppose that $\mathbf{v}_t^\top \nabla f(\mathbf{Q}_t)\mathbf{v}_t < 0$. In this case we have from the definition of $\mu_t$ that $\mu_t \in [0, 1]$. In particular, if $\mathbf{Q}_t \succ 0$ we have that $\mu_t \in [0, 1)$ and so, by the definition of $\mathbf{Q}_{t+1}$ it follows that $\mathbf{Q}_{t+1} \succ 0$ as well.

For the second case in which $\mathbf{v}_t^\top \nabla f(\mathbf{Q}_t)\mathbf{v}_t > 0$, using the assumption that $\mathbf{Q}_t \succ 0$ and Observation 2, we first note that $\mu_t < 0$, and so $(1 - \mu_t)\mathbf{Q}_t \succ 0$. Denoting $\mathbf{A}_t = (1 - \mu_t)\mathbf{Q}_t$, we make the following observations:

$$
\begin{aligned}
\mathbf{A}_t + \mu_t\mathbf{v}_t\mathbf{v}_t^\top \succ 0 &\iff \mathbf{A}_t^{1/2}\left(\mathbf{I} + \mu_t\mathbf{A}_t^{-1/2}\mathbf{v}_t\mathbf{v}_t^\top\mathbf{A}_t^{-1/2}\right)\mathbf{A}_t^{1/2} \succ 0 \\
&\iff \mathbf{I} + \mu_t\mathbf{A}_t^{-1/2}\mathbf{v}_t\mathbf{v}_t^\top\mathbf{A}_t^{-1/2} \succ 0 \\
&\iff \lambda_{\min}\left(\mathbf{I} + \mu_t\mathbf{A}_t^{-1/2}\mathbf{v}_t\mathbf{v}_t^\top\mathbf{A}_t^{-1/2}\right) > 0 \\
&\iff 1 + \mu_t\lambda_{\max}\left(\mathbf{A}_t^{-1/2}\mathbf{v}_t\mathbf{v}_t^\top\mathbf{A}_t^{-1/2}\right) > 0 \qquad (\mu_t < 0) \\
&\iff 1 + \mu_t\mathbf{v}_t^\top\mathbf{A}_t^{-1}\mathbf{v}_t > 0 \iff 1 + \frac{\mu_t}{1 - \mu_t}\mathbf{v}_t^\top\mathbf{Q}_t^{-1}\mathbf{v}_t > 0.
\end{aligned}
$$

Note that $\frac{\mu_t}{1-\mu_t} = -\frac{\mathbf{v}_t^\top \nabla f(\mathbf{Q}_t)\mathbf{v}_t}{(\mathbf{v}_t^\top\mathbf{Q}_t^{-1}\mathbf{v}_t)^2}$, and so

$$
\begin{aligned}
1 + \frac{\mu_t}{1 - \mu_t}\mathbf{v}_t^\top\mathbf{Q}_t^{-1}\mathbf{v}_t &= 1 - \frac{\mathbf{v}_t^\top \nabla f(\mathbf{Q}_t)\mathbf{v}_t}{\mathbf{v}_t^\top\mathbf{Q}_t^{-1}\mathbf{v}_t} \\
&= 1 + \frac{1}{\mathbf{v}_t^\top\mathbf{Q}_t^{-1}\mathbf{v}_t}\left(\frac{p}{n}\sum_{i=1}^n \frac{\mathbf{v}_t^\top\mathbf{Q}_t^{-1}\mathbf{x}_i\mathbf{x}_i^\top\mathbf{Q}_t^{-1}\mathbf{v}_t}{\mathbf{x}_i^\top\mathbf{Q}_t^{-1}\mathbf{x}_i} - \mathbf{v}_t^\top\mathbf{Q}_t^{-1}\mathbf{v}_t\right) \\
&= \frac{1}{\mathbf{v}_t^\top\mathbf{Q}_t^{-1}\mathbf{v}_t}\frac{p}{n}\sum_{i=1}^n \frac{\mathbf{v}_t^\top\mathbf{Q}_t^{-1}\mathbf{x}_i\mathbf{x}_i^\top\mathbf{Q}_t^{-1}\mathbf{v}_t}{\mathbf{x}_i^\top\mathbf{Q}_t^{-1}\mathbf{x}_i} = \frac{1}{\mathbf{v}_t^\top\mathbf{Q}_t^{-1}\mathbf{v}_t}\frac{p}{n}\sum_{i=1}^n \frac{(\mathbf{v}_t^\top\mathbf{Q}_t^{-1}\mathbf{x}_i)^2}{\mathbf{x}_i^\top\mathbf{Q}_t^{-1}\mathbf{x}_i}.
\end{aligned}
$$

The RHS is non-negative and it is equal to zero if and only if for all $i = 1, \ldots, n$ it holds that $\mathbf{x}_i \perp \mathbf{Q}_t^{-1}\mathbf{v}_t$, however this implies that there exists a subspace of dimension $p - 1$ containing all $n$ data-points, which is in contrast to Assumption 1. $\qquad\square$

## A.2 Proof of Theorem 2

*Proof.* We first focus on the efficient implementation of the AFW step (Eq. (6)) and then show how it relates to computing the GAFW step (Eq. (7)). Then, we discuss the efficient implementation of the FW step (Eq. (5)).

Fix some iteration $t$ of Algorithm 1. We reduce the computation of $\mathbf{v}_t$ which satisfies (6) to three approximate leading eigenvalue computations. Fix some $\tilde{\beta} \in [0, 1)$ to be determined later. First, we compute a unit vector $\mathbf{u}$ such that $\mathbf{u}^\top (\nabla f(\mathbf{Q}_t))^2 \mathbf{u} \geq (1 - \tilde{\beta})^2 \lambda_1((\nabla f(\mathbf{Q}_t))^2) = (1 - \tilde{\beta})^2 \|\nabla f(\mathbf{Q}_t)\|_2^2$, and we compute $C = \sqrt{\mathbf{u}^\top (\nabla f(\mathbf{Q}_t))^2 \mathbf{u}}$. Next we define the matrices $\mathbf{M}_+ = (1 - \tilde{\beta})^{-1} C \mathbf{I} + \nabla f(\mathbf{Q}_t)$, $\mathbf{M}_- = (1 - \tilde{\beta})^{-1} C \mathbf{I} - \nabla f(\mathbf{Q}_t)$. Note that by definition, both $\mathbf{M}_+, \mathbf{M}_-$ are positive semidefinite. Next we compute unit vectors $\mathbf{u}_+, \mathbf{u}_-$, which are approximate leading eigenvectors, with a factor $(1 - \tilde{\beta})$ multiplicative approximation, of the matrices $\mathbf{M}_+, \mathbf{M}_-$, respectively. That is,

$$\mathbf{u}_+^\top \mathbf{M}_+ \mathbf{u}_+ \geq (1 - \tilde{\beta}) \lambda_1(\mathbf{M}_+) = (1 - \tilde{\beta}) \left( (1 - \tilde{\beta})^{-1} C + \lambda_1(\nabla f(\mathbf{Q}_t)) \right)$$

$$\mathbf{u}_-^\top \mathbf{M}_- \mathbf{u}_- \geq (1 - \tilde{\beta}) \lambda_1(\mathbf{M}_-) = (1 - \tilde{\beta}) \left( (1 - \tilde{\beta})^{-1} C - \lambda_p(\nabla f(\mathbf{Q}_t)) \right),$$

which using the definition of $\mathbf{M}_+, \mathbf{M}_-$ implies that

$$\mathbf{u}_+^\top \nabla f(\mathbf{Q}_t) \mathbf{u}_+ \geq (1 - \tilde{\beta}) \lambda_1(\nabla f(\mathbf{Q}_t)) - \tilde{\beta}(1 - \tilde{\beta})^{-1} C$$

$$-\mathbf{u}_-^\top \nabla f(\mathbf{Q}_t) \mathbf{u}_- \geq -(1 - \tilde{\beta}) \lambda_p(\nabla f(\mathbf{Q}_t)) - \tilde{\beta}(1 - \tilde{\beta})^{-1} C.$$

Thus,

$$\max\{\mathbf{u}_+^\top \nabla f(\mathbf{Q}_t) \mathbf{u}_+, -\mathbf{u}_-^\top \nabla f(\mathbf{Q}_t) \mathbf{u}_-\} \geq (1 - \tilde{\beta}) \|\nabla f(\mathbf{Q}_t)\|_2 - \tilde{\beta}(1 - \tilde{\beta})^{-1} C$$

$$\underset{(a)}{\geq} \left( 1 - \tilde{\beta} - \frac{\tilde{\beta}}{1 - \tilde{\beta}} \right) \|\nabla f(\mathbf{Q}_t)\|_2$$

$$\geq \frac{1 - 3\tilde{\beta}}{1 - \tilde{\beta}} \|\nabla f(\mathbf{Q}_t)\|_2,$$

where in (a) we have used the fact that by definition $C \leq \|\nabla f(\mathbf{Q}_t)\|_2$.

Thus, setting $\tilde{\beta}$ such that $\frac{1 - 3\tilde{\beta}}{1 - \tilde{\beta}} \geq 1 - \beta$ and taking $\mathbf{v}_t = \sqrt{p} \cdot \arg\max_{\mathbf{w} \in \{\mathbf{u}_-, \mathbf{u}_+\}} |\mathbf{w}^\top \nabla f(\mathbf{Q}_t) \mathbf{w}|$, we obtain the result required by (6).

Since for a universal constant $\beta$, $\tilde{\beta}$ as defined above is also a universal constant, computing each approximated eigenvector $\mathbf{u}, \mathbf{u}_+, \mathbf{u}_-$ using the well-known *power method* with random initialization requires $O\left(\log \frac{p}{\delta}\right)$ iterations, where $\delta$ is the desired failure probability, see for instance [8] (Theorem A.1). Each such iteration of the power method requires to compute a matrix-vector product with a matrix of the form $c\mathbf{I} \pm \nabla f(\mathbf{Q}_t)$, where $c$ is a given scalar. Thus, it remains to detail how to compute fast matrix-vector products with the gradient $\nabla f(\mathbf{Q}_t)$. Fix some vector $\mathbf{v}$. It holds that

$$\nabla f(\mathbf{Q}_t)\mathbf{v} = -\frac{p}{n} \sum_{i=1}^{n} \frac{\mathbf{Q}_t^{-1} \mathbf{x}_i \mathbf{x}_i^\top \mathbf{Q}_t^{-1} \mathbf{v}}{\mathbf{x}_i^\top \mathbf{Q}_t^{-1} \mathbf{x}_i} + \mathbf{Q}_t^{-1} \mathbf{v}.$$

Note that if the vectors $\mathbf{y}_{t,i} := \mathbf{Q}_t^{-1} \mathbf{x}_i, i = 1, \ldots, n$ are stored explicitly in memory, and recalling that $\mathbf{Q}_t^{-1}$ is also computed and stored explicitly, computing $\nabla f(\mathbf{Q}_t)\mathbf{v}$ requires overall only $O(np + p^2)$ time. It thus remains to show how given the vectors $\mathbf{y}_{t,i}, i = 1, \ldots, n$, and the vector $\mathbf{v}_t$, we can quickly compute the new vectors $\mathbf{y}_{t+1,i} := \mathbf{Q}_{t+1}^{-1} \mathbf{x}_i, i = 1, \ldots, n$.

Using the Sherman-Morrison formula we have that,

$$\mathbf{Q}_{t+1}^{-1} = ((1 - \mu_t)\mathbf{Q}_t + \mu_t \mathbf{v}_t \mathbf{v}_t^\top)^{-1} = \frac{1}{1 - \mu_t} \left( \mathbf{Q}_t^{-1} - \gamma_t \frac{\mathbf{Q}_t^{-1} \mathbf{v}_t \mathbf{v}_t^\top \mathbf{Q}_t^{-1}}{1 + \gamma_t \mathbf{v}_t^\top \mathbf{Q}_t^{-1} \mathbf{v}_t} \right),$$

where we recall that $\gamma_t = \frac{\mu_t}{1 - \mu_t}$.

Thus, for all $i = 1, \ldots, n$ we have that,

$$\mathbf{y}_{t+1,i} = \mathbf{Q}_{t+1}^{-1}\mathbf{x}_i = \frac{1}{1-\mu_t}\mathbf{y}_{t,i} - \frac{\gamma_t}{1-\mu_t}\frac{\mathbf{Q}_t^{-1}\mathbf{v}_t\mathbf{v}_t^\top\mathbf{y}_{t,i}}{1+\gamma_t\mathbf{v}_t^\top\mathbf{Q}_t^{-1}\mathbf{v}_t}.$$

Thus, after computing $\mathbf{Q}_t^{-1}\mathbf{v}_t$, we can indeed compute $\mathbf{y}_{t+1,i}$ form $\mathbf{y}_{t,i}$ in $O(p)$ time, and overall $O(np + p^2)$ to compute all vectors $\mathbf{y}_{t+1,i}, i = 1, \ldots, n$.

We now turn to discuss the efficient computation of the GAFW step in Eq. (7). As discussed in Section 2, computing $\mathbf{v}_t$ in this case amounts to finding a unit vector $\mathbf{u}$ such that $|\mathbf{u}^\top\mathbf{Q}^{1/2}\nabla f(\mathbf{Q}_t)\mathbf{Q}^{1/2}\mathbf{u}| \geq (1-\beta)\|\mathbf{Q}_t^{1/2}\nabla f(\mathbf{Q}_t)\mathbf{Q}_t^{1/2}\|_2$, and returning $\mathbf{v}_t = \sqrt{p}\frac{\mathbf{Q}_t^{1/2}\mathbf{u}}{\|\mathbf{Q}_t^{1/2}\mathbf{u}\|}$. Thus, given the matrix $\mathbf{Q}_t^{1/2}$ explicitly, computing such unit vector $\mathbf{u}$, and then the corresponding vector $\mathbf{v}_t$, could be carried out in time $\tilde{O}(p^2 + np)$ using the same reasoning as that in the computation of the AFW step. In particular, each matrix-vector product of the form $\mathbf{Q}_t^{1/2}\nabla f(\mathbf{Q}_t)\mathbf{Q}_t^{1/2}\mathbf{v}$ could be carried out in $O(p^2 + np)$ times as explained above. Thus, using additional $O(p^3)$ to explicitly compute the matrix $\mathbf{Q}_t^{1/2}$, we obtain the result of the theorem for the GAFW updates.

Finally, for the FW updates (Eq. (5)), we compute the constant $C$ and define the matrix $\mathbf{M}_-$ as before, but this time we compute the vector $\mathbf{u}_-$ (there is no need for $\mathbf{u}_+$ in the standard FW step) so that $\mathbf{u}_-^\top\mathbf{M}_-\mathbf{u}_- \geq (1-\hat{\beta})\lambda_1(\mathbf{M}_-)$, for some $\hat{\beta} \in [0,1)$, to be determined shortly. Similarly to the analysis for AFW, this yields,

$$\mathbf{u}_-^\top\nabla f(\mathbf{Q}_t)\mathbf{u}_- \leq (1-\hat{\beta})\lambda_p(\nabla f(\mathbf{Q}_t)) + \hat{\beta}(1-\tilde{\beta})^{-1}C$$
$$\leq (1-\hat{\beta})\lambda_p(\nabla f(\mathbf{Q}_t)) + \hat{\beta}(1-\tilde{\beta})^{-1}\|\nabla f(\mathbf{Q}_t)\|_2.$$

Thus, choosing for instance $\tilde{\beta} = 1/2$ and $\hat{\beta} = \frac{|\lambda_p(\nabla f(\mathbf{Q}_t))|}{3\|\nabla f(\mathbf{Q}_t)\|_2}\beta$, we indeed have that $\mathbf{u}_-^\top\nabla f(\mathbf{Q}_t)\mathbf{u}_- \leq (1-\beta)\lambda_p(\nabla f(\mathbf{Q}_t))$, and so, returning $\mathbf{v}_t = \sqrt{p}\mathbf{u}_-$ indeed satisfies (5).

Using standard results for the power method (see for instance Theorem A.1 in [8]), as discusses above, computing such $\mathbf{u}_-$ with failure probability at most $\delta$ requires $O\left(\frac{\|\nabla f(\mathbf{Q}_t)\|_2}{|\lambda_p(\nabla f(\mathbf{Q}_t))|\beta}\log(p\delta^{-1})\right)$ matrix-vector products. This could be improved to only $O\left(\sqrt{\frac{\|\nabla f(\mathbf{Q}_t)\|_2}{|\lambda_p(\nabla f(\mathbf{Q}_t))|\beta}}\log(p\delta^{-1})\right)$ matrix-vector products using the faster Lanczos method [12, 23, 21]. □

## B Proofs Missing from Section 3

### B.1 Proof of Observation 1

*Proof.* We first establish that $\lambda_{\min}(\nabla f(\mathbf{Q})) \leq 0$ and $\lambda_{\min}(\nabla f(\mathbf{Q})) = 0$ if and only if $\mathbf{Q} = \mathbf{Q}^*$. Then, the observation follows since $\mathbf{Q} \succ 0$ and $\mathbf{Q}\nabla f(\mathbf{Q})\mathbf{Q} = \mathbf{Q} - \frac{p}{n}\sum_{i=1}^n \frac{\mathbf{x}_i\mathbf{x}_i^\top}{\mathbf{x}_i^\top\mathbf{Q}^{-1}\mathbf{x}_i}$.

A straight-forward calculation shows that for any $\mathbf{Q} \in \mathcal{S}_{p+}$ it holds that $\langle\mathbf{Q}, \nabla f(\mathbf{Q})\rangle = 0$. Writing the eigen-decomposition of $\nabla f(\mathbf{Q})$ as $\nabla f(\mathbf{Q}) = \sum_{i=1}^p \lambda_i\mathbf{u}_i\mathbf{u}_i^\top$, this implies that

$$0 = \langle\mathbf{Q}, \nabla f(\mathbf{Q})\rangle = \sum_{i=1}^p \lambda_i\mathbf{u}_i^\top\mathbf{Q}\mathbf{u}_i.$$

Since $\mathbf{Q} \in \mathcal{S}_{p+}$ implies that for all $i = 1, \ldots, p$, $\mathbf{u}_i^\top\mathbf{Q}\mathbf{u}_i > 0$, it follows that $\lambda_p \leq 0$, and moreover, $\lambda_p = 0$ if and only if $\lambda_i = 0$ for all $i = 1, \ldots, p$, meaning $\nabla f(\mathbf{Q}) = 0$, which holds if and only if $\mathbf{Q}$ satisfies (1), i.e., $\mathbf{Q} = \mathbf{Q}^*$. □

### B.2 Proof of Theorem 3

*Proof.* Fix $\tilde{\epsilon} > 0$. Corollary 1 establishes that for all variants and for all $t \geq T(\tilde{\epsilon})$, it holds that $\min_{\tau=0,\ldots,t-1}|L_\tau| \leq \tilde{\epsilon}$. Let us denote by $t^*$ an index of such iteration for which it holds that $|L_{t^*}| \leq \tilde{\epsilon}$.

Let us begin with FW steps. By definition we have that

$$|L_t| = \frac{|\mathbf{v}_t^\top \nabla f(\mathbf{Q}_t)\mathbf{v}_t|}{\mathbf{v}_t^\top \mathbf{Q}_t^{-1}\mathbf{v}_\tau} = \frac{-\mathbf{v}_t^\top \nabla f(\mathbf{Q}_t)\mathbf{v}_\tau}{\mathbf{v}_t^\top \mathbf{Q}_t^{-1}\mathbf{v}_t} \geq \frac{-p(1-\beta)\lambda_{\min}(\nabla f(\mathbf{Q}_t))}{p\lambda_{\min}^{-1}(\mathbf{Q}_t)}$$

$$= -(1-\beta)\lambda_{\min}(\mathbf{Q}_t)\lambda_{\min}\left(\mathbf{Q}_t^{-1}\left(\mathbf{Q}_t - \frac{p}{n}\sum_{i=1}^{n}\frac{\mathbf{x}_i\mathbf{x}_i^\top}{\mathbf{x}_i^\top \mathbf{Q}_t^{-1}\mathbf{x}_i}\right)\mathbf{Q}_t^{-1}\right)$$

$$\geq -(1-\beta)\lambda_{\min}(\mathbf{Q}_t)\lambda_{\max}^{-2}(\mathbf{Q}_t)\lambda_{\min}\left(\mathbf{Q}_t - \frac{p}{n}\sum_{i=1}^{n}\frac{\mathbf{x}_i\mathbf{x}_i^\top}{\mathbf{x}_i^\top \mathbf{Q}_t^{-1}\mathbf{x}_i}\right).$$

Thus, for $\tilde{\epsilon} = \tilde{\epsilon}_{\mathrm{FW}}$ and $t = t^*$ we indeed have that (8) holds.

We now turn to prove (9) holds for AFW and GAFW updates. For AFW updates, using similar arguments as before, it holds that

$$|L_t| = \frac{|\mathbf{v}_t^\top \nabla f(\mathbf{Q}_t)\mathbf{v}_t|}{\mathbf{v}_t^\top \mathbf{Q}_t^{-1}\mathbf{v}_t} \geq (1-\beta)\lambda_{\min}(\mathbf{Q}_t)\left\|\mathbf{Q}_t^{-1}\left(\mathbf{Q}_t - \frac{p}{n}\sum_{i=1}^{n}\frac{\mathbf{x}_i\mathbf{x}_i^\top}{\mathbf{x}_i^\top \mathbf{Q}_t^{-1}\mathbf{x}_i}\right)\mathbf{Q}_t^{-1}\right\|_2$$

$$\geq (1-\beta)\lambda_{\min}(\mathbf{Q}_t)\lambda_{\max}^{-2}(\mathbf{Q}_t)\left\|\mathbf{Q}_t - \frac{p}{n}\sum_{i=1}^{n}\frac{\mathbf{x}_i\mathbf{x}_i^\top}{\mathbf{x}_i^\top \mathbf{Q}_t^{-1}\mathbf{x}_i}\right\|_2.$$

Thus, for $\tilde{\epsilon} = \tilde{\epsilon}_{\mathrm{AFW}}$ and $t = t^*$ we indeed have that (9) holds for AFW.

We turn to prove that (9) holds for GAFW updates. According to the update rule we have that,

$$|L_t| = \frac{|\mathbf{v}_t^\top \nabla f(\mathbf{Q}_t)\mathbf{v}_t|}{\mathbf{v}_t^\top \mathbf{Q}_t^{-1}\mathbf{v}_t} \geq (1-\beta)\|\mathbf{Q}_t^{1/2}\nabla f(\mathbf{Q}_t)\mathbf{Q}_t^{1/2}\|_2$$

$$= (1-\beta)\left\|\mathbf{Q}_t^{-1/2}\left(\mathbf{Q}_t - \frac{p}{n}\sum_{i=1}^{n}\frac{\mathbf{x}_i\mathbf{x}_i^\top}{\mathbf{x}_i^\top \mathbf{Q}_t^{-1}\mathbf{x}_i}\right)\mathbf{Q}_t^{-1/2}\right\|_2$$

$$\geq (1-\beta)\lambda_{\max}^{-1}(\mathbf{Q}_t)\left\|\mathbf{Q}_t - \frac{p}{n}\sum_{i=1}^{n}\frac{\mathbf{x}_i\mathbf{x}_i^\top}{\mathbf{x}_i^\top \mathbf{Q}_t^{-1}\mathbf{x}_i}\right\|_2.$$

Thus, for $\tilde{\epsilon} = \tilde{\epsilon}_{\mathrm{GAFW}}$ and $t = t^*$ we have that (9) holds for GAFW. $\qquad\square$

## C  Proofs Missing from Section 4

### C.1  Proof of Theorem 5

Before we can prove Theorem 5 we need to establish several auxiliary results.

**Lemma 4.** *Under Assumption 2, there exist a constant $\alpha > 0$ such that for any pair of unit vectors* $\mathbf{u}, \mathbf{v}$ *in* $\mathbb{R}^p$*, it holds that* $\frac{1}{n}\sum_{i=1}^{n}(\mathbf{u}^\top \mathbf{x}_i)^2(\mathbf{v}^\top \mathbf{x}_i)^2 \geq \alpha$.

*Proof.* Let $\mathbf{u}^*, \mathbf{v}^*$ be the optimal solutions to the optimization problem:

$$\min_{\mathbf{u}, \mathbf{v}: \|\mathbf{u}\| = \|\mathbf{v}\| = 1} \frac{1}{n}\sum_{i=1}^{n}(\mathbf{u}^\top \mathbf{x}_i)^2(\mathbf{v}^\top \mathbf{x}_i)^2 = \alpha,$$

and suppose by way of contradiction that $\alpha = 0$.

Thus, for all $i = 1, \ldots, n$, $\mathbf{x}_i \perp \mathbf{u}^*$ or $\mathbf{x}_i \perp \mathbf{v}^*$, which in particular implies that there exists $\mathcal{S} \subseteq \{\mathbf{x}_1, \ldots, \mathbf{x}_n\}$ such that $|\mathcal{S}| \geq n/2$ and either $\mathbf{u}^*$ or $\mathbf{v}^*$ are orthogonal to all vectors in $\mathcal{S}$, which implies orthogonality to $\mathrm{span}(\mathcal{S})$. However, according to Assumption 2, $\mathrm{span}(\mathcal{S}) = \mathbb{R}^p$, and thus we arrive at a contradiction. $\qquad\square$

**Lemma 5.** *Suppose Assumption 2 holds. Consider the function*

$$g(t) := f(\mathbf{Z}^{\frac{1}{2}}\exp{(t\mathbf{W})}\mathbf{Z}^{\frac{1}{2}}),$$

*for some $\mathbf{Z} \succ 0$, and $\mathbf{W} \in \mathbb{S}^p$ such that $Tr(\mathbf{W}) = 0$ and $\|\mathbf{W}\|_1 = 1$. Then,*

$$\forall t: \quad g''(t) \geq \left(\frac{\lambda_{\min}(\mathbf{Z})}{\lambda_{\max}(\mathbf{Z})}\right)^2 p^{-1} \exp(-4t)\alpha,$$

*where $\alpha > 0$ is the constant implied by Lemma 4.*

*Proof.* Write the eigen-decomposition of $\mathbf{W}$ as $\mathbf{W} = \sum_{j=1}^p \lambda_j \mathbf{u}_j \mathbf{u}_j^\top$. Calculations give

$$\begin{aligned} g(t) &= f(\mathbf{Z}^{\frac{1}{2}} \exp(t\mathbf{W}) \mathbf{Z}^{\frac{1}{2}}) \\ &= \frac{p}{n} \sum_{i=1}^n \log\left(\sum_{j=1}^p \exp(-\lambda_j t)(\mathbf{x}_i^\top \mathbf{Z}^{-\frac{1}{2}} \mathbf{u}_j)^2\right) + t \cdot \mathrm{Tr}(\mathbf{W}) + \log\det(\mathbf{Z}). \end{aligned}$$

Thus,

$$g'(t) = -\frac{p}{n} \sum_{i=1}^n \frac{\sum_{j=1}^p \lambda_j \exp(-\lambda_j t)(\mathbf{x}_i^\top \mathbf{Z}^{-\frac{1}{2}} \mathbf{u}_j)^2}{\sum_{j=1}^p \exp(-\lambda_j t)(\mathbf{x}_i^\top \mathbf{Z}^{-\frac{1}{2}} \mathbf{u}_j)^2} + \mathrm{Tr}(\mathbf{W}),$$

which implies that,

$$\begin{aligned} g''(t) = \frac{p}{n} \sum_{i=1}^n &\frac{\left(\sum_{j=1}^p \lambda_j^2 \exp(-\lambda_j t)(\mathbf{x}_i^\top \mathbf{Z}^{-\frac{1}{2}} \mathbf{u}_j)^2\right)\left(\sum_{j=1}^p \exp(-\lambda_j t)(\mathbf{x}_i^\top \mathbf{Z}^{-\frac{1}{2}} \mathbf{u}_j)^2\right)}{(\sum_{j=1}^p \exp(-\lambda_j t)(\mathbf{x}_i^\top \mathbf{Z}^{-\frac{1}{2}} \mathbf{u}_j)^2)^2} \\ &- \frac{\left(\sum_{j=1}^p \lambda_j \exp(-\lambda_j t)(\mathbf{x}_i^\top \mathbf{Z}^{-\frac{1}{2}} \mathbf{u}_j)^2\right)^2}{(\sum_{j=1}^p \exp(-\lambda_j t)(\mathbf{x}_i^\top \mathbf{Z}^{-\frac{1}{2}} \mathbf{u}_j)^2)^2}. \end{aligned}$$

Let us introduce the notation $C_{ij} = \exp(-\lambda_j t)(\mathbf{x}_i^\top \mathbf{Z}^{-\frac{1}{2}} \mathbf{u}_j)^2$ for all $i \in \{1, \ldots, n\}, j \in \{1, \ldots, p\}$. We have that

$$\begin{aligned} g''(t) &= \frac{p}{n} \sum_{i=1}^n \frac{\left(\sum_{j=1}^p \lambda_j^2 C_{ij}\right)\left(\sum_{j=1}^p C_{ij}\right) - \left(\sum_{j=1}^p \lambda_j C_{ij}\right)^2}{(\sum_{j=1}^p C_{ij})^2} \\ &= \frac{p}{n} \sum_{i=1}^n \frac{\sum_{j=1}^p \sum_{k=1}^p \frac{\lambda_j^2 + \lambda_k^2}{2} C_{ij} C_{ik} - \sum_{j=1}^p \sum_{k=1}^p \lambda_j \lambda_k C_{ij} C_{ik}}{(\sum_{j=1}^p C_{ij})^2} \\ &= \frac{p}{n} \sum_{i=1}^n \frac{\sum_{j=1}^p \sum_{k=1}^p \frac{(\lambda_j - \lambda_k)^2}{2} C_{ij} C_{ik}}{(\sum_{j=1}^p C_{ij})^2}. \end{aligned}$$

Re-arranging we get,

$$g''(t) = \sum_{j=1}^p \sum_{k=1}^p (\lambda_j - \lambda_k)^2 \frac{p}{n} \sum_{i=1}^n \frac{C_{ij} C_{ik}}{2(\sum_{l=1}^p C_{il})^2}.$$

Let us fix some $i, j$, and recall that $C_{ij} = \exp(-\lambda_j t)(\mathbf{x}_i^\top \mathbf{Z}^{-\frac{1}{2}} \mathbf{u}_j)^2$. Since $\|\mathbf{W}\|_2 \leq \|\mathbf{W}\|_1 = 1$, we have that

$$\exp(-t)\lambda_{\max}^{-1}(\mathbf{Z})\left(\mathbf{x}_i^\top \frac{\mathbf{Z}^{-1/2} \mathbf{u}_j}{\|\mathbf{Z}^{-1/2} \mathbf{u}_j\|}\right)^2 \leq C_{ij} \leq \exp(t)\lambda_{\min}^{-1}(\mathbf{Z}).$$

Thus, for any $j, k = 1, \ldots, p$ we have that,

$$
\frac{p}{n} \sum_{i=1}^{n} \frac{C_{ij} C_{ik}}{2(\sum_{l=1}^{p} C_{il})^2} \geq \frac{p}{n} \sum_{i=1}^{n} \frac{\lambda_{\max}^{-2}(\mathbf{Z}) \exp(-2t) \left( \mathbf{x}_i^{\top} \frac{\mathbf{Z}^{-\frac{1}{2}} \mathbf{u}_j}{\|\mathbf{Z}^{-\frac{1}{2}} \mathbf{u}_j\|} \right)^2 \left( \mathbf{x}_i^{\top} \frac{\mathbf{Z}^{-\frac{1}{2}} \mathbf{u}_k}{\|\mathbf{Z}^{-\frac{1}{2}} \mathbf{u}_k\|} \right)^2}{2(p \lambda_{\min}^{-1}(\mathbf{Z}) \exp(t))^2}
$$

$$
\geq \frac{\left( \frac{\lambda_{\min}(\mathbf{Z})}{\lambda_{\max}(\mathbf{Z})} \right)^2 \exp(-4t)}{2p} \frac{1}{n} \sum_{i=1}^{n} \left( \mathbf{x}_i^{\top} \frac{\mathbf{Z}^{-\frac{1}{2}} \mathbf{u}_j}{\|\mathbf{Z}^{-\frac{1}{2}} \mathbf{u}_j\|} \right)^2 \left( \mathbf{x}_i^{\top} \frac{\mathbf{Z}^{-\frac{1}{2}} \mathbf{u}_k}{\|\mathbf{Z}^{-\frac{1}{2}} \mathbf{u}_k\|} \right)^2
$$

$$
\geq \frac{\left( \frac{\lambda_{\min}(\mathbf{Z})}{\lambda_{\max}(\mathbf{Z})} \right)^2 \exp(-4t)\alpha}{2p},
$$

where the last inequality is due to Lemma 4.

Introducing the notation $\delta(\mathbf{Z}) = \frac{\lambda_{\max}(\mathbf{Z})}{\lambda_{\min}(\mathbf{Z})}$, we finally we get,

$$
g''(t) \geq \frac{\exp(-4t)\alpha}{2p\delta^2(\mathbf{Z})} \sum_{j=1}^{p} \sum_{k=1}^{p} (\lambda_j - \lambda_k)^2 = \frac{\exp(-4t)\alpha}{2p\delta(\mathbf{Z})^2} \sum_{j=1}^{p} \sum_{k=1}^{p} (\lambda_j{}^2 - 2\lambda_j \lambda_k + \lambda_k{}^2)
$$

$$
= \frac{\exp(-4t)\alpha}{2p\delta(\mathbf{Z})^2} (2p\|\mathbf{W}\|_F^2 - 2\mathrm{Tr}(\mathbf{W})^2) \underset{(a)}{=} \delta(\mathbf{Z})^{-2} \exp(-4t)\alpha \|\mathbf{W}\|_F^2
$$

$$
\geq p^{-1}\delta(\mathbf{Z})^{-2} \exp(-4t)\alpha \|\mathbf{W}\|_1^2 \underset{(b)}{=} p^{-1}\delta(\mathbf{Z})^{-2} \exp(-4t)\alpha,
$$

where in (a) we have used the assumption that $\mathrm{Tr}(\mathbf{W}) = 0$, and in (b) we have used the assumption that $\|\mathbf{W}\|_1 = 1$.

$\square$

**Lemma 6.** *Suppose Assumption 2 holds. Let $\mathbf{Z} \in \mathbb{S}_{++}$ such that $\det(\mathbf{Z}) = 1$, and let $\mathbf{Z}^* \succ 0$ be a minimizer of $f(\cdot)$ such that $\det(\mathbf{Z}^*) = 1$. It holds that*

$$
\|\mathbf{Z}^{1/2} \nabla f(\mathbf{Z}) \mathbf{Z}^{1/2}\|_2^2 \geq 2\kappa^{-2}(\mathbf{Z}) p^{-1} \exp(-4D(\mathbf{Z}, \mathbf{Z}^*))\alpha \cdot (f(\mathbf{Z}) - f(\mathbf{Z}^*)), \qquad (14)
$$

*where $\kappa(\mathbf{Z}) := \lambda_{\max}(\mathbf{Z})/\lambda_{\min}(\mathbf{Z})$, $D(\mathbf{Z}, \mathbf{Z}^*) := \|\log(\mathbf{Z}^{-\frac{1}{2}} \mathbf{Z}^* \mathbf{Z}^{-\frac{1}{2}})\|_1$, and $\alpha > 0$ is the constant implied by Lemma 4.*

*Proof.* Let $\mathbf{W} = \frac{\log(\mathbf{Z}^{-1/2} \mathbf{Z}^* \mathbf{Z}^{-1/2})}{D}$, where $D = D(\mathbf{Z}, \mathbf{Z}^*) = \|\log(\mathbf{Z}^{-1/2} \mathbf{Z}^* \mathbf{Z}^{-1/2})\|_1$. Note that

$$
\mathrm{Tr}\left( \log(\mathbf{Z}^{-1/2} \mathbf{Z}^* \mathbf{Z}^{-1/2}) \right) = \sum_{i=1}^{p} \lambda_i \left( \log(\mathbf{Z}^{-1/2} \mathbf{Z}^* \mathbf{Z}^{-1/2}) \right)
$$

$$
= \sum_{i=1}^{p} \log \left( \lambda_i(\mathbf{Z}^{-1/2} \mathbf{Z}^* \mathbf{Z}^{-1/2}) \right)
$$

$$
= \log \left( \Pi_{i=1}^{p} \lambda_i(\mathbf{Z}^{-1/2} \mathbf{Z}^* \mathbf{Z}^{-1/2}) \right)
$$

$$
= \log \det(\mathbf{Z}^{-1/2} \mathbf{Z}^* \mathbf{Z}^{-1/2})
$$

$$
= \log(\det(\mathbf{Z}^{-1/2}) \cdot \det(\mathbf{Z}^*) \cdot \det(\mathbf{Z}^{-1/2}))
$$

$$
= \log(\det^{-1}(\mathbf{Z}) \cdot \det(\mathbf{Z}^*)) = 0,
$$

where the last equality is due to the fact that $\det(\mathbf{Z}) = \det(\mathbf{Z}^*) = 1$.

Note also that by definition $\|\mathbf{W}\|_1 = 1$. Denote the function

$$
g(t) = f(\mathbf{Z}^{\frac{1}{2}} \exp(t\mathbf{W}) \mathbf{Z}^{\frac{1}{2}}),
$$

and note that $g(0) = f(\mathbf{Z})$, $g(D) = f(\mathbf{Z}^*)$.

Thus, using Lemma 5, and denoting the constant $A = \kappa^{-2}(\mathbf{Z})p^{-1}\exp(-4D(\mathbf{Z}, \mathbf{Z}^*))\alpha$, we have that,

$$\forall t \in [0, D]: \quad g''(t) \geq A.$$

Integrating on both sides we have that,

$$\forall t \in [0, D]: \quad g'(t) \geq g'(0) + At.$$

Integrating again we have,

$$\forall t \in [0, D]: \quad g(t) - g(0) \geq t \cdot g'(0) + \frac{At^2}{2}.$$

It holds that

$$
\begin{aligned}
g'(0) &= \langle \nabla f(\mathbf{Z}^{1/2}\exp(t\mathbf{W})\mathbf{Z}^{1/2}), \mathbf{Z}^{1/2}\mathbf{W}\exp(t\mathbf{W})\mathbf{Z}^{1/2}\rangle|_{t=0} \\
&= \langle \nabla f(\mathbf{Z}), \mathbf{Z}^{1/2}\mathbf{W}\mathbf{Z}^{1/2}\rangle = \langle \mathbf{Z}^{1/2}\nabla f(\mathbf{Z})\mathbf{Z}^{1/2}, \mathbf{W}\rangle \\
&\geq -\|\mathbf{Z}^{1/2}\nabla f(\mathbf{Z})\mathbf{Z}^{1/2}\|_2,
\end{aligned}
$$

where the last inequality is due to Hölder's inequality and the fact that $||\mathbf{W}||_1 = 1$.

Plugging-back we have that,

$$\forall t \in [0, D]: \quad g(t) - g(0) \geq -t\|\mathbf{Z}^{1/2}\nabla f(\mathbf{Z})\mathbf{Z}^{1/2}\|_2 + \frac{At^2}{2}.$$

The right hand side obtains its minimal value for $t^* = \frac{\|\mathbf{Z}^{1/2}\nabla f(\mathbf{Z})\mathbf{Z}^{1/2}\|_2}{A}$, which yields,

$$\forall t \in [0, D]: \quad g(0) - g(t) \leq \frac{\|\mathbf{Z}^{1/2}\nabla f(\mathbf{Z})\mathbf{Z}^{1/2}\|_2^2}{2A},$$

and specifically,

$$f(\mathbf{Z}) - f(\mathbf{Z}^*) = g(0) - g(D) \leq \frac{\|\mathbf{Z}^{1/2}\nabla f(\mathbf{Z})\mathbf{Z}^{1/2}\|_2^2}{2A},$$

which concludes the proof. $\qquad\square$

We can now prove Theorem 5

*Proof of Theorem 5.* Given $\mathbf{Q}_0$ and $\mathbf{Q}$, define their unit-determinant normalizations $\mathbf{Z}_0 := \det^{-1/p}(\mathbf{Q}_0)\mathbf{Q}_0$, $\mathbf{Z} := \det^{-1/p}(\mathbf{Q})\mathbf{Q}$. Denote also $\mathbf{Z}^* = \det^{-1/p}(\mathbf{Q}^*)\mathbf{Q}^*$ (recall that since $f(\cdot)$ is invariant to scaling, $\mathbf{Z}^*$ also minimizes $f(\cdot)$). From Lemma 6 we have that,

$$\|\mathbf{Z}^{1/2}\nabla f(\mathbf{Z})\mathbf{Z}^{1/2}\|_2^2 \geq 2\kappa^{-2}(\mathbf{Z})p^{-1}\exp(-4D(\mathbf{Z}, \mathbf{Z}^*))\alpha \cdot (f(\mathbf{Z}) - f(\mathbf{Z}^*)),$$

where $\kappa(\mathbf{Z}) := \lambda_{\max}(\mathbf{Z})/\lambda_{\min}(\mathbf{Z})$, $D(\mathbf{Z}, \mathbf{Z}^*) := ||\log(\mathbf{Z}^{-\frac{1}{2}}\mathbf{Z}^*\mathbf{Z}^{-\frac{1}{2}})||_1$, and $\alpha > 0$ is the constant implied by Lemma 4.

Using the fact that the functions $f(\mathbf{Z})$, $\kappa(\mathbf{Z})$ are invariant to scaling of $\mathbf{Z}$, and so is the matrix $\mathbf{Z}^{1/2}\nabla f(\mathbf{Z})\mathbf{Z}^{1/2}$, we have that

$$\|\mathbf{Q}^{1/2}\nabla f(\mathbf{Q})\mathbf{Q}^{1/2}\|_2^2 \geq 2\kappa^{-2}(\mathbf{Q})p^{-1}\exp(-4D(\mathbf{Z}, \mathbf{Z}^*))\alpha \cdot (f(\mathbf{Q}) - f(\mathbf{Q}^*)).$$

Recall that according to Lemma 1, for every $\mathbf{Q}$ in the level set $\{\mathbf{Q} \in \mathcal{S}_p \mid f(\mathbf{Q}) \leq f(\mathbf{Q}_0)\}$, $\lambda_{\min}(\mathbf{Q}) > \lambda$, for some $\lambda > 0$. This further implies that

$$D(\mathbf{Z}, \mathbf{Z}^*) = \|\log(\mathbf{Z}^{-\frac{1}{2}}\mathbf{Z}^*\mathbf{Z}^{-\frac{1}{2}})\|_1 = \|\log(\det^{1/p}(\mathbf{Q})\det^{-1/p}(\mathbf{Q}^*)\mathbf{Q}^{-\frac{1}{2}}\mathbf{Q}^*\mathbf{Q}^{-\frac{1}{2}})\|_1$$

is upper-bounded by some constant on the level-set $\{\mathbf{M} \in \mathcal{S}_p \mid f(\mathbf{M}) \leq f(\mathbf{Q}_0)\}$, which yields the corollary. $\qquad\square$

## C.2 Proof of Theorem 4

*Proof.* Let us first consider the case of AFW steps. From Lemma 3 we have that out of $t$ iterations Algorithm 1 has executed, the maximum number of iterations $\tau$ in which it holds that $L_\tau^2 > 1$, is at most $\lceil 4(f(\mathbf{Q}_0) - f(\mathbf{Q}^*)) \rceil = \lceil 4h_0 \rceil$, where we recall that $L_\tau := \frac{\mathbf{v}_\tau^\top \nabla f(\mathbf{Q}_\tau) \mathbf{v}_\tau}{\mathbf{v}_\tau^\top \mathbf{Q}_\tau^{-1} \mathbf{v}_\tau}$. On any other iteration $\tau$ we have that,

$$
\begin{aligned}
f(\mathbf{Q}_{\tau+1}) - f(\mathbf{Q}_\tau) &\leq -\frac{1}{4}L_\tau^2 \underset{(a)}{\leq} -\frac{(1-\beta)^2 p^2 \|\nabla f(\mathbf{Q}_\tau)\|_2^2}{4(\mathbf{v}_\tau^\top \mathbf{Q}_\tau^{-1} \mathbf{v}_\tau)^2} \\
&= -\frac{(1-\beta)^2 p^2 \|\mathbf{Q}_\tau^{-1/2} \mathbf{Q}_\tau^{1/2} \nabla f(\mathbf{Q}_\tau) \mathbf{Q}_\tau^{1/2} \mathbf{Q}_\tau^{-1/2}\|_2^2}{4(\mathbf{v}_\tau^\top \mathbf{Q}_\tau^{-1} \mathbf{v}_\tau)^2} \\
&\leq -\frac{(1-\beta)^2 p^2 \lambda_{\max}^{-2}(\mathbf{Q}_\tau) \|\mathbf{Q}_\tau^{1/2} \nabla f(\mathbf{Q}_\tau) \mathbf{Q}_\tau^{1/2}\|_2^2}{4p^2 \lambda_{\min}^{-2}(\mathbf{Q}_\tau)} \\
&\underset{(b)}{\leq} -\frac{(1-\beta)^2 \kappa_0^{-2} \rho \left( f(\mathbf{Q}_\tau) - f(\mathbf{Q}^*) \right)}{4},
\end{aligned}
$$

where (a) follows from the definition of $L_\tau$ and the AFW update step (Eq. (6)), and (b) follows from Theorem 5.

Rearranging, we have that

$$
h_{\tau+1} \leq h_\tau \left( 1 - \frac{(1-\beta)^2}{4} \kappa_0^{-2} \rho \right) \leq h_\tau \exp\left( -\frac{(1-\beta)^2}{4} \kappa_0^{-2} \rho \right).
$$

Thus, we can conclude that,

$$
h_t \leq h_0 \exp\left( -\frac{(1-\beta)^2}{4} \kappa_0^{-2} \rho \left( t - \lceil 4h_0 \rceil \right) \right).
$$

The result for GAFW steps follows from the same reasoning, only now using the GAFW update step (Eq. (7)) which gives the improved bound:

$$
-\frac{1}{4}L_\tau^2 \leq -\frac{(1-\beta)^2}{4} \|\mathbf{Q}_\tau^{1/2} \nabla f(\mathbf{Q}_\tau) \mathbf{Q}_\tau^{1/2}\|_2^2.
$$

$\square$