# OpenReview forum: "Frank-Wolfe-based Algorithms for Approximating Tyler's M-estimator"
_NeurIPS.cc/2022/Conference — NeurIPS 2022 Accept_

### Official Review · Reviewer_Q7uf · 2022-06-15

**Rating:** 4
**Confidence:** 3
**Soundness:** 2 fair
**Presentation:** 2 fair
**Contribution:** 2 fair

**Summary:**

The paper is aimed at approximating Tyler's M-estimator by proposing a series of new methods, which, compared with the previous fixed-point approach, are cheaper in the complexity per iteration. They propose three kinds of Frank-Wolfe methods, following the framework of standard FW, away-step FW, and geodesic FW, and prove their sublinear convergence rates. Furthermore, when a mild additional assumption also holds, they can prove their linear convergence. In my view, the most significant improvement over the previous fix-point method is that they avoid the expensive matrix inverse and propose new methods which can be applied to large-scale problems. Finally, the improvements are demonstrated by some small experiments.

**Questions:**

1. What is the relationship of the PL parameter and the data? It shows in Table 1 without a thorough explanation. Will it be super small in the worst case? I suggest the author show that it is not bad in a certain proper way.
2. I am familiar with the Frank-Wolfe method but I still suggest authors add more literature review about the Frank-Wolfe method, and explain the idea and general framework of the Frank-Wolfe method separately (not in Algorithm 1). I believe the current layout, especially the way this paper introduces Eq. (5-7), might be very confusing for readers.
3. The release of the codes can help make the paper more convincing.
4. Can FW methods really exhibit linear convergence in real experiments? It is not obvious in the current experiment.
5. Does FW methods still have advantages over the fixed-point methods in real-world datasets instead of artificial data? I observe in Table 1 that the complexity is highly related to the condition number of Q, so the dataset might play an important role too.
6. There is inequality between the nuclear norm and the Frobenius norm. I feel changing the norm in Theorem 3 might make it clearer.

Smaller issues:
Line 24: the definition of N(L) is not clear. I think the number of points in a subspace should be infinite? Does that mean the number of points in {x_i}?
Line 52: FPI is not defined
Line 69: Omega is not defined
Line 327: Typo in reference

**Limitations:**

I think the authors do well in this aspect.

**Strengths And Weaknesses:**

Strength:
I am not an expert in Tyler's M-estimator, so more literature review about the other algorithms in dealing with the problem can help me a lot. But in general, I think this paper has a significant contribution in decreasing the per iteration complexity, compared with the fixed-point method, and the idea of proving the convergence of FW for this nonsmooth nonconvex problem is also interesting and novel.

Weakness:
1. In general, the structure and logic of this paper are clear but there are still some issues of writing, which make it hard for a person without relevant knowledge to understand.
2. The dependence of the PL parameter on the data is not studied enough. I doubt whether it can be extremely bad in the worst case. If so, the comparison in Table 1 can be very misleading and the improvement over the fixed-point method will be not enough for publication.
3. The experiment results don't show the priority of the three FW methods enough. I think a figure of showing the convergence results (linear convergence and sublinear convergence of FW methods) should also be necessary. And I am also interested in the complexity of getting the eigenvector of the smallest eigenvalue for standard FW methods.

---

> ### Author Response · Authors · 2022-07-31
> **Reply**
>
> Dear reviewer,
>
> First we address the weakness you raise:
> 1: We give all relevant references in the paper and we basically give most of the background needed for Tyler's estimator in the paper. The Frank-Wolfe method is well known and well studied and we cannot review it in the paper but point to the standard references. Note that Section 2 clearly details all variants and how they relate to classical Frank-Wolfe variants. Note also that two reviewers found the presentation to be good.
>
> 2. As we write, the dependence of the PL parameter and the data does not admit a simple form and that is why we do not explicitly state it. This is very common in convex optimization and often the PL parameter has complex dependence on the data but one can establish that it exists (bounded away from zero). Studying it numerically for instance is beyond our interest which is mostly focused on rigorous theoretical analysis, and beyond the scope of this paper. This does not make Table 1 misleading since the parameter clearly appear in the bounds and should be understood that in certain cases it can be quite bad. Note, that while the linear convergence rate may perhaps be bad, our sublinear rates are still interesting and allow for fast approximation of Tyler's estimator without dependence on this parameter.
>
> 3. The complexity of computing the smallest eigenvalue of FW appears in the proof of Theorem 2 in the appendix. Regarding experiments: as we wrote above, our main interest in this work is on novel theoretical approach and analysis for approximating Tyler's estimator and expanding the theory of Frank-Wolfe methods. Comprehensive numerical tests, are beyond our interest. In the experiments we did we tested on the settings considered in [28], since for these [28] showed that the Tyler's estimator is indeed interesting to you use which makes these settings of interest. Indeed the FW and AFW do not perform too well in these cases, but it is what it is. We do see that GAFW has very good performance since it achieves good approximation error even before the baseline FPI completes a single iteration.
>
> Answers to questions:
> 1+2: see comments above.
>
> 3. We will include the code in final submission.
>
> 4+5. This is a theoretical paper and our main interest in in obtaining novel and state-of-the-art complexity bounds, as well, as extending our understanding of optimization methods. We touch upon numerical experiments but this is not our main interest. We used these two settings, because in [28] it was shown that in these settings Tyler's estimator is indeed significantly superior to the sample covariance, and so these settings are meaningful and natural to test. Indeed in principle the constant in the exponent of the linear rate, which depends on the data in a very complicated way, can be quite bad, and this is probably the reason that FW and AFW variants do not seem to converge linearly in the experiments.
>
> 6. Unfortunately, we do not understand this comment.
>
> Smaller issues:
> 1. the definition of N(L) is not clear: it is the number of *data points* that lie inside the subspace L, note the data is finite.
>  2. FPI is defined in Definition 1
> 3. Omega is the standard complexity lower bound notation
>
> Finally, since you clearly write that ''...in general, I think this paper has a significant contribution in decreasing the per iteration complexity, compared with the fixed-point method, and the idea of proving the convergence of FW for this nonsmooth nonconvex problem is also interesting and novel'' we cannot understand how a score of REJECT could be acceptable. It is fine if you are perhaps not very familiar with the related research and literature and if you do not like the paper much, but given that you understand that this paper contains significant and novel ideas w.r.t. both computing Tyler's estimator and the theory of Frank-Wolfe, and that you did not find a technical flaw, we do not feel this is very professional. We sincerely ask you to reconsider your score, or at least lower your confidence score.

---

> > ### Comment · Reviewer_Q7uf · 2022-08-08
> > **Reply**
> >
> > Thank the authors for the reply.
> >
> > The additional experimental results clearly show the linear convergence of the FW methods the authors proposed. Although the computational experiments are still too simple, these new results at least make the authors’ claim more convincing. Because of this, I will raise the score a little (3->4) but I insist that the experiments should be presented with more details and comparisons, even if the authors believe it is a theoretical paper.
> >
> > I also insist that the presentation of this paper needs more improvements, and emphasizes more key contributions. Besides, the constant of the linear convergence is also essential. Even if revealing the theoretical bounds is difficult, the computational experiments should give readers more feeling about it. Linear convergence is common in many methods but usually the bad constant prevents it from being widely used, such as subgradient methods for LP.

---

> > > ### Author Response · Authors · 2022-08-08
> > > **Response**
> > >
> > > Thank you for your response and for reconsidering your score. We would like to further comment on the issues raised in your last response.
> > >
> > > Experiments:
> > > 1. You write `` I insist that the experiments should be presented with more details and comparisons''. Could you please explain what you exactly mean by more details and comparisons? We believe we have given all the relevant details and we are not sure what more comparisons are you referring to.
> > >
> > > 2. You also write ''the computational experiments should give readers more feeling about it [the constant of linear convergence]''. We would like to emphasize again that while the linear rate is definitely nice, even our sublinear rates, which do not depend on such constant, are highly novel in terms of running times implications for computing the TME (since they apply linear time iterations, while enjoying first-order like convergence rates), and thus we do not feel they should be discarded just because there are also linear rate results.
> > > Regarding the constant in linear rate: again, we have used the standard examples from the excellent survey on Tyler's estimator [28]. Our results show that one of our method could indeed be notably faster than fixed-point iterations. We sincerely do not feel coming up with more artificial examples to demonstrate the linear rate will be highly beneficial
> > >
> > > Presentation:
> > > The key contributions of this paper are: i. the first methods for provable approximation of Tyler's estimator that has linear runtime per-iteration (as opposed to super-linear of the fixed iterations method), and in particular overall linear runtime for well conditioned instances and when the target accuracy epsilon is not very low, and ii. novel Frank-Wolfe methods (for example, a geodesic variant of Frank-Wolfe, which we believe is a novel contribution by itself, or the use of our adaptive and simple step-sizes) for solving to global optimality and nonsmooth and nonconvex problem of notable interest, which is highly likely to lead to more advances on Frank-Wolfe variants for stylized nonconvex problems with novel running times (recall Frank-Wolfe usually solve to global optimality only smooth and convex problems).
> > > Both of this we believe are clearly highlighted throughout the introduction.
> > > Could you please expand on how do you think the current presentation is lacking in explaining these contributions?

---

### Official Review · Reviewer_6Bnc · 2022-06-17

**Rating:** 6
**Confidence:** 4
**Soundness:** 3 good
**Presentation:** 3 good
**Contribution:** 3 good

**Summary:**

This paper discusses a new way of computing the Tyler covariance estimator, which is the solution $Q$ to the fixed point relation
$\frac{p}{n} \sum_{i=1}^n \frac{x_ix_i^T}{x_i^TQ^{\*-1}x_i} = Q^{*}$.
Here, $p$ is the feature dimension and $n$ is the number of samples.
The usual method is to evaluate this iteratively, with $Q_k^{-1}$ on the left hand side and $Q_{k+1}$ on the right hand side. There are two main computational issues here: inverting $Q$, which is $O(p^3)$ and evaluating the matrix/vector multiplications $O(np^2)$ at each iteration.

This paper tries instead to find this fixed point using a Frank-Wolfe method, where at each iteration, they first compute a $1-\beta$-approximate LMO (a low rank matrix), and then merge it to $Q$ using a carefully designed merge parameter $\mu_k$. The idea is that FW is solving the (nonconvex) problem of minimizing $f(Q) = \frac{p}{n}\sum_{i=1}^n \log(x_i^TQ^{-1}x_i) + \log\det(Q)$ over the constraint $Q\succ 0, tr(Q) = p$. By using lanczos methods, the approximate LMOs can be found of order $O(p^2+np)$ per iteration, over $O(1/k^2)$ iterations to get to $\epsilon$ error.

**Questions:**

See previous box

**Strengths And Weaknesses:**

 - The approach is very clever, and though is heavily FW based, is not a trivial extension at all. Especially, the choice of $\mu_k$ seems to have this method deviate from the usual rates ($O(1/k)$ not $O(1/k^2)$).

 - After taking a "careful skim" of the paper, I do not see any red flags in terms of proofs or weirdly magical steps. However, there are some things I think should be clarified (see below).

 - First, it is confusing to me that the per iteration rate for FW and AFW are so different. I believe AFW has actually two steps, one for the min LMO and the other for the max LMO. So it puzzles me that it doesn't have about the same per-iteration rate (if not worse) than FW. Besides, AFW also has the memory overhead of holding onto all past atoms--how is this avoided in this implementation?

 - The idea of keeping both $Q$ and $Q^{-1}$ through the Sherman Morrison Woodbury formula is very interesting, but I wonder if numerical error will accumulate because of this. Can the authors comment?

 - In general, if there is to be proofs in the main paper (and not the appendix) I think it is better if each step is discussed more intuitively, with the expectation that the reader will basically follow all the main points without a ton of effort. Maybe the proof to Lemma 3 can be amended to be more clear.

 - I would also like to see more discussion on the construction of the $\mu_k$ sequence, since that seems to be a significant novelty (usually, we pick $\mu_k = 2/(2+k)$ and obtain an $O(1/k)$ iterations rate to $\epsilon$ error, and we use a very different proof technique.)

 - the numerical results are not strong. Though the FPI has a large per-iteration rate, I'm not sure Fig. 1 supports this new approach well. While I do think there is enough theoretical novelty in this paper that we don't need SOTA numerical results, I do think a better motivating example would make the paper much stronger.

If all of these points are addressed clearly by the authors, I may raise my score.

---

> ### Author Response · Authors · 2022-07-31
> **Reply**
>
> Dear reviewer,
>
> Thank you for finding our work interesting and novel, we appreciate it.
>
> We now answer the various issues:
> 1. FW/AFW: AFW is expected to converge faster since it uses more sophisticated updates. Note that in our setting, and very different from AFW for polytopes (which is usually the setting studied for AFW), we can efficiently implement the ''away steps'' implicitly without storing all previous atoms, that is one of the beauties of this variant in our spectrahedron setting. Technically, since under Assumption 1, the optimum lies in the interior it follows that computing the away-step becomes just an eigenvector problem which is efficient to solve.
>
> 2. Use of Sherman-Morisson: this is a very common primitive in numerical algorithms to the best of our knowledge and we are not aware of stability issues. Admittedly though, our main interest in this work is mostly on theoretical analysis and establishing the convergence results in principle.
>
> 3. Lemma 3: we will revise the proof and attempt to make it more accessible. We believe this is indeed important since it is the main technical step in the analysis, and of interest to those who wish to understand the very basic idea of our analysis.
>
> 4. step-size: The 2/(2+k) step-size is common for convex objectives. Here our analysis of Frank-Wolfe is for nonconvex objective (convergence to stationary points) which usually uses different step-sizes, see for instance  [18]. We shall clarify this in the final version.
>
> 5. Numerical experiments: Indeed, our main focus in on novel theoretical approach and analysis, and extensive experiments are beyond our interest here. The reason we used these settings is that in [28] it was shown that for these settings Tyler's estimator is indeed superior to the sample covariance and thus interesting. We did not want to come up with artificial settings that will give us nice graphs but which are pointless since they do not capture really interesting cases. Nevertheless, we feel that GAFW indeed shows promising performance: it achieves non-trivial approximation even before FPI completes a single iteration! In particular, on the left panel, by the time FPI completes the first iteration, GAFW has already obtained the minimal value which FPI achieves only in its final iteration. Moreover, in the final version we shall add a graph in which the Y-axis is in log-scale which makes it clearer that the GAFW variant is indeed notably faster than FPI throughout the run.

---

> > ### Comment · Reviewer_6Bnc · 2022-08-03
> > **reply**
> >
> > Thanks for the reply.
> >
> > 1. Indeed, it looks like the AFW that you implemented looks fundamentally different than the AFW presented in Lacoste-Julien 2015. Given that you give convergence rates for this new method (and don't borrow from his), that seems kosher, but maybe consider a name change, or specify this is a different method more clearly.
> >
> > 2. Fair enough. This point feels minor, since it doesn't take up much room in the main text.
> >
> > 4. I agree, and I commented in a positive way: I am more curious on the intuition behind this choice of step size, and how it fits your specific problem / gives specific rates. This intuition would help the broader Frank-Wolfe community in other instances, too.
> >
> > 3 and 5: looking forward to seeing the updates.

---

> > > ### Author Response · Authors · 2022-08-08
> > > **response**
> > >
> > > Dear Reviewer 6Bnc,
> > >
> > > In case you have missed it, we have updated our submission and add an appendix with additional numerical results as promised (these maybe integrated into the main text in the final version). Per your original review, if your main concerns have been answered, will you consider raising you score as you suggested?
> > >
> > > We will be very happy to answer additional questions.

---

### Official Review · Reviewer_6Ljb · 2022-07-11

**Rating:** 7
**Confidence:** 5
**Soundness:** 3 good
**Presentation:** 3 good
**Contribution:** 3 good

**Summary:**

The paper proposes a Frank-Wolfe algorithm for
approximating a well-known covariance matrix estimator.
The estimator is known to minimize a particular non-convex,
locally gradient dominated function (as shown in Theorem 5)
over the set of positive definite matrices with a fixed trace,
i.e., a convex set which is a neither open nor closed.
These are atypical settings for a Frank-Wolfe algorithm,
nevertheless the paper establishes strong convergence results
for the original Frank-Wolfe algorithm and two variants with away
steps, e.g., linear convergence in function value
for the away-step variants
for a non-convex function over a non-polyhedral domain.
The convergence is for a quantity similar to the dual gap
for the original algorithm.

The computational results are poor: only one setting, wrong data displayed (accuracy in a help function of a method instead of accuracy for the original problem independent of method of solution, measuring cost via theoretical bounds instead of actual performance), making a claim even contradicting the graph.

**Questions:**

None.

**Limitations:**

Not applicable.

**Strengths And Weaknesses:**

The theoretical results are excellent for the theory of Frank-Wolfe
algorithms with correct proofs, even providing detailed discussion of theoretical upper bounds for costs of each operation.  A slight weakness is the presentation of results in Table 1: the last two columns display different measures of rates without this being clearly indicated.

Since the problem is specific to appplication of covariance estimators,
it would have been good to address performance for the original
covariance estimating problem, which it is meant to solve.
For this an accuracy measure for covariance matrix estimators should
be studied independent of the method of solution.

In the computational experiments,
such a measure ought to be displayed on the vertical axis,
instead of the function value of questionable practical relevance.

On the horizontal axis, the authors stated aim was a measure
independent of implementation issues, a worthy goal, but unfortunately
the chosen "estimated runtime" is the best worst-case known bound
on operation cost, which is heavily biased against
algorithms with high worst-case computational cost.
For example, such a measure would show the ellipsoid methpd
as far faster for linear programming than the simplex method,
despite the latter being faster in practice.
Due this error it is not possible to deduce acutal performance from the graphs,
e.g., the baseline FPI might have outperformed all the other algorithms.

The claim in Line 300 that GAFW is significantly faster than FPI
is contradicting to the graphs, even if we ignore the errors
discussed above.  FPI has the same performance on the right and nnly
slightly worse on the left, so the difference is insignificant.

Overall the computational results are worthless, and likely to cause only misinterpretations.

---

> ### Author Response · Authors · 2022-07-31
> **Reply**
>
> Dear reviewer,
>
> Thank you very much for you high appreciation of our theoretical results, we truly appreciate it.
>
> Issues:
> 1. We shall add a clarification to Table 1.
> 2. Measure of convergence: these are the measures of performance that arise naturally from the analysis and this is why we study them. They could be translated to other measures but it makes sense to us to report convergence on these because this is what come out of the analysis. For instance, our measure of convergence for the AFW and GAFW methods in Theorem 3 give approximation in spectral norm w.r.t. the exact Tyler estimator. This seems highly natural to us, and is also very similar to the measure of convergence in the previous excellent work [6] which studied rigouros convergence guarantees for the fixed-point iterations.
>
> Regarding experiments, we do not quite share you feelings and let us explain:
> 1. We use two settings not one (corrupted gaussian and heavy tailed distribution) since these were shown in [28] to be cases in which Tyler's estimator indeed makes sense and is considerably better that the sample covariance. It is important for us not only to display nice graphs but such that concern settings of true potential interest.
> 2. Measure of convergence: there are several possible measures and all are equivalent in the sense that Tyler's estimator is their minimizer. We therefor do not think that the objective function, of which Tyler's estimator is the only minimizer, is a poor choice of measure. In the final version we will add an additional measure such as the distance from the exact Tyler's estimator (in spectral norm for instance). The graphs look very similar.
> 3. Measuring runtime: if we wanted to measure actual runtime that would have required us to implement on our own a specialized state-of-the-art eigenvector method such as Lanczos for computing an eigenvector corresponding to the largest (in magnitude) eigenvalue, that works with our particular efficient updates (see Section 2.1), and to run quite high-dimenisional and time-costly simulations for the differences between the methods to be clear. This is something that is beyond the scope of our current research that is mostly dedicated to novel theoretical results.
> We feel that the estimated runtime issue you raise is much milder than you suggest: we simply make the assumption that a dense matrix - dense vector product takes O(n^2) time, where the matrix is nxn, and a vector-vector product takes O(n) time. We feel this is quite reasonable.
> 4. ''The claim in Line 300 that GAFW is significantly faster than FPI is contradicting to the graphs'': there seems to be some misunderstanding here. The blue dots in the graph mark the iterations of FPI. So, we can see that GAFW makes significant progress and achieves reasonable approximation errors before FPI has even made a single iteration! This is what we meant, and we will clearly clarify it in the final version.  Note also that in the left graph, by the time that FPI has completed a single iteration, GAFW has approximately reached the value FPI would achieve only on its last iteration. In the final version we shall also add a graph which plots the Y-axis in log scale which makes it much easier to see that GAFW is indeed notably faster.

---

> > ### Author Response · Authors · 2022-08-10
> > **Reply**
> >
> > Dear Reviewer 6Ljb,
> >
> > Did our response and additional graphs helped with your concerns regarding experiments?
> >
> > We would be happy to answer additional questions/issues.

---

### Official Review · Reviewer_6ubv · 2022-07-11

**Rating:** 5
**Confidence:** 3
**Soundness:** 3 good
**Presentation:** 1 poor
**Contribution:** 2 fair

**Summary:**

The submission considers a Frank-Wolfe based algorithm for approximating Tyler's M-Estimator. Overall, main benefits of Frank-Wolfe based algorithms for this problem is the lower computational complexity per iteration. Compared to the standard fixed-point iteration, the computation complexity is improved from $O(np^2)$ to $O(np)$. Three variants of Frank-Wolfe algorithms are proposed, and sublinear convergence guarantees are provided for them. With a slightly milder assumption, the paper shows linear convergence.

**Questions:**

- It is hard to understand why Assumption 2 brings a major improvement (from sublinear to linear), because it sounds very natural for all i.i.d. data. It would be better to come up with more directly related assumption for the improvement.

- The linear convergence results could have been more stressed, instead of filling one full page (page 7) with a bunch of non-informative algebras. Please discuss the relation to the PL condition in a more detail, starting from the definition of it, in more detail. It could have made the paper more interesting.

**Limitations:**

I do not see any negative societal impact.

**Strengths And Weaknesses:**

Overall, presentation of the paper is not very good. I do not see major benefits of presenting three variants in parallel. The authors could first have elaborated on the basic FW algorithm and develop the most basic ideas first, e.g., how to establish sublinear convergence and what are the main technical challenges. Then, the authors could have proposed two more variants (AFW, GAFW) that enjoys slightly better convergence, and faster convergence with slightly milder conditions. In its current form, it is very hard to understand what are the major technical challenges. Maybe more importantly, I wonder why the Frank-Wolfe approach is interesting especially when the scope of the paper is very specific to one objective function. There could be other approaches that lead to similar computational benefits other than FW.

---

> ### Author Response · Authors · 2022-07-31
> **Reply**
>
> Dear reviewer,
>
> First we answer your questions:
> 1. Indeed the beauty of it is that Assumption 2 is so mild that generally we shall have a linear rate almost generically. However, the constant in the exponent of this rate might in general be quite bad and depends on the data in quite a complicated way (e.g., the AFW in our experiments does not seem to exhibit a linear rate), and this is why the sublinear rates are interesting - they still allow for fast approximation (when the desired error epsilon is not to small) while not being dependent to this constant.
>
> 2. The PL condition and linear convergence is discussed on page 8 and defined in line 262. This inequality is by now very well known in the continuous optimization literature (see the relevant references [22, 17, 9]) and the use of it to obtain linear rates, and so we do not go into much detail. For this we have the proof with all supporting Lemmas
>
> Regarding writing style: This is a theoretical paper and our main contribution is a novel use of the classical Frank-Wolfe method to solve an important NONCONVEX NONSMOOTH optimization problem for covariance estimation. With this respect, Lemma 7 for instance is not a bunch of ''non-informative algebras'' but the main technical argument that allows us to establish convergence and as a theory paper this is important in our opinion.
>
> As to your additional point regarding use of Frank-Wolfe: FW here is the platform that allows us to break the super-linear runtime per iteration of the FPI method and to obtain linear runtime per iteration. This is highly interesting also in the border context of FW, since we solve to global optimality a nonconvex nonsmooth problem, while FW can guarantee global convergence usually only for convex and smooth problems. We think that if you can think of another method to obtain such a result, you should definitely write a paper about it.
>
> We understand that you did not like the way the paper is written and that you are perhaps not familiar with this line of research, but note that two reviewers found the presentation good, and we, as experienced scientific authors, are entitled for our personal taste which sould not be fully aligned with the preferences of every possible reviewer, and to us the presentation seems very natural.
>
> Finally, our paper gives highly novel results both w.r.t. the theory of Frank-Wolfe and computing Tyler's estimator, and as such, we believe it would be interesting to many in the NeurIPS community. We have worked hard for months on this project, and we truly believe our theoretical results are clearly above bar. In all honesty, we do not think that the issues you raised, and in particular your individual preferences of writing style, should be an acceptable cause for recommending rejection. You did not find any technical flaw or indicated that our results are not novel or uninteresting in some way. We ask you to seriously reconsider your score, or at least your confidence level.

---

> > ### Comment · Reviewer_6ubv · 2022-08-08
> > **Reply to Rebuttal**
> >
> > Thanks to authors for the response. The writing I suggested is just one option, and I did not mean to criticize the authors' taste. However, I still feel that the current presentation is suboptimal for several reasons.
> >
> > I think the paper conveys two messages:
> >
> > (1) FW can be a solution to Tyler's M-estimator (but why is it interesting?)
> >
> > (2) FW (for Tyler's M-estimator) can be faster with some modified schemes, even though it is not an order-wise improvement.
> >
> > For (1), the benchmark (fixed-point iteration) does not seem very strong. Fixed-point iteration is such a simple algorithm which is appealing in practice, but if we only care to improve the order-wise computational complexity, I doubt that no previous work has studied this. Positioning of the paper in literature does not seem clear in the current version. It could be due to my lack of background on related work, but I feel that this should have been sufficiently addressed.
> >
> > Also, it seems that Frank-Wolfe algorithm has been studied for non-convex objectives (quick google search shows me a few other references [1,2]). Given that Q* is the only stationary point of the objective function, which is nice, but then it is not clear why we should study FW only for Tyler's M-estimator separately. In this respect, I think that the current version fails in positioning of the work as well.
> >
> > [1] Stochastic Frank-Wolfe Methods for Nonconvex Optimization
> >
> > [2] Decentralized Frank-Wolfe Algorithm for Convex and Non-convex Problems
> >
> > On the technical side, related to the positioning, it is not clear what are the main challenge and key contributions. In almost all optimization papers, conclusions are like "it converges". I would like to see more: why some textbook derivations don't work, and thus what new ideas are required. But in the current version, other than explaining Sherman-Morrison formula, and some seemingly textbook derivations in Page 7, I do not see what are key technical contributions. The improvement of $O(p)$ merely comes from the matrix inversion update using Sherman-Morrison formula, which does not sound surprising. Yes, this is my personal opinion and evaluation, but I also think that my perception could have been different if presented better (though I don't know what would have been optimal).
> >
> >
> > For (2), I also would like to see why and how the modified schemes bring the improvement. But other than the mechanism explanation, I do not see the intuition behind the ideas or what are additional challenges if applying modified schemes. I thought it might have been clearer if faster methods are presented after the ideas of standard FW part become clear (again, I don't know what is optimal).
> >
> >
> > Section 4 is also not super clear about what brings the linear rate improvement, which step breaks down if we do not have Assumption 2, what is the order of some important constants, etc. Frankly, it seems a bit suspicious because Assumption 2 is such a general statement which holds for almost all distributions with well-conditioned convariances, and as pointed out in other reviews, constants seem quite bad.

---

> > > ### Author Response · Authors · 2022-08-09
> > > **Response**
> > >
> > > Thank you Reviewer 6ubv for your additional comments. We answer them below.
> > >
> > > 1. Misunderstanding of main messages: you seem to repeat the same mistake in both points. Our FW-based methods give the first algorithms algorithms for TME that can run in linear time! Of course this could be very significant!
> > >
> > > 2. You write: '' (fixed-point iteration) does not seem very strong'': Fixed point iterations (FPI) are the method of choice for computing Tyler's estimator and you can see variants of it in basically all papers we gave in the literature review. We are not aware of any competing methods. In particular note FPI seems very strong: it has linear convergence, and it requires only ''simple'' operations like a single matrix inversion per iteration and multiplying a matrix with each data-point on each iteration. Only due to the very unique structure of the rank-one updates of Frank-Wolfe, that we are able to break these ''bottlenecks'' and obtain methods with linear runtime per-iteration, and as we write explicitly, we are not aware of any previous method with running times as ours.
> > >
> > > 3. We do not think that a sentence like ''I doubt that no previous work has studied this. Positioning of the paper in literature does not seem clear in the current version'' is a professional review style. We have gave literature review. If you know of works that we missed but are important please state so, otherwise such a comment is not nor respectful. Note that we cite many recent works and to the best of our knowledge we are up to speed with relevant methods.
> > >
> > > 3. You write: ''Also, it seems that Frank-Wolfe algorithm has been studied for non-convex objectives'':  We split our answer to 3 parts:
> > >
> > > 3.a. These methods your mention are for smooth problems. Here the problem is not smooth. Designing an efficient stochastic FW variant is interesting but is beyond the scope of our work and not-trivial since our analysis heavily relies on the fact that the algorithm is a descent method - it reduces the function value on each iteration hence remaining in the initial level set, which is difficult to argue for stochastic methods. This is a good question for future research and such a result will most definitely build on our arguments.
> > >
> > > 3.b. One of our main interests in this paper is obtaining faster algorithms for Tyler's estimator. This is achieved not by using Frank-Wolfe as a ''black box'', but noticing that the specific structure of rank-one updates used in Frank-Wolfe for the spectrahedron could lead to significant acceleration of runtime of other operations: such as matrix inversion (via Sherman Morrison) and stylized highly efficient eigenvector computation (which for instance avoid computing the gradient explicitly which is expensive). Theses allow for iteration cost of O(np) which would otherwise cost (np^2) like in FPI - please refer to Section 2.1.
> > > So, one of the key observations here is that there is a  ''very good match'' between the updates of FW and the algebraic structure of the specific problem which makes this such an interesting combination and as a consequence allows for faster runtime.
> > >
> > > 3.c. We develop unique techniques in the landscape of Frank-Wolfe methods including: 1. an adaptive step-size procedure that avoids the need to tune parameters which are not really known (like the minimial eigenvalue of a matrix on the initial level-set) and 2. a *geodesic FW variant* which we think is a very interesting contribution.
> > >
> > > 4. You write: ''On the technical side...it is not clear what are the main challenge and key contributions...and some seemingly textbook derivations in Page 7, I do not see what are key technical contributions'': As we wrote. We solve a nonsmooth and nonconvex problem via FW. Of course this is a major challenge, since FW is not adapted to handling nonsmooth objectives! The derivations on Page 7 are by no means ''textbook derivations'', but they are the heart of our technical analysis, and even while appearing simple, it is this unique analysis that considers the specific algebraic structure of the problem at hand, that allows us to obtain our novel convergence results, which as we emphasize again, are non-standard for first-order methods as a whole, and FW methods in particular, since problem is non-convex and nonsmooth.
> > >
> > > 5. You write ''Section 4...which step breaks down if we do not have Assumption 2...'':
> > >
> > > 5a. We are limited in space and we do not think it is important to discuss what breaks in analysis if the assumption does not hold. For this please refer to the proof.
> > >
> > > 5b. We cannot give a compact representation of the linear conv. constant or how bad it is. This is something that is quite common in linear rates for first-order methods and we kindly refer you the references we provided to see that it is a standard thing. It is common that such constants may have complex dependence on the data.
> > >
> > > We kindly ask you again to reconsider your score. We are very happy to answer additional concerns.

---

> > > > ### Comment · Reviewer_6ubv · 2022-08-09
> > > > **Fair Point**
> > > >
> > > > Okay, it is quite surprising that no work has developed faster algorithms than FPI for Tyler's M-estimator. After some literature survey, I could find some work in sparse settings, but it seems that there really doesn't exist algorithm that runs faster than $O(p^3)$. The paper is more about finding a Tyler's M-estimator in time less than $O(p^3)$, and having this scope in mind, now the paper makes a lot more sense. I increase my score from 3 to 5.
> > > >
> > > > It would have been a lot more helpful for non-knowledgeable (on Tyler's M-estimator) readers if the scope was made much clear in the beginning, and if the paper provides a more thorough discussion on existing work for solving Tyler's M-estimator problem.
> > > >
> > > > I think it is a slight overclaim that the paper also contributes to theories of FW (and AFW or GAFW) for non-convex non-smooth problems, because the current analysis is very specific to and applies only to the Tyler's estimator problem. It is not clear how to extend the analysis beyond this specific problem.
> > > >
> > > > I still cannot agree with the technical writing style in this paper, but I agree that this is totally up to authors' freedom.
> > > >
> > > >
> > > > Besides, I wonder whether other GD based algorithms such as Projected (sub)-gradient descent can achieve the same performance. There is only one stationary point which is a global optimum, and so it is kind of expected that any GD based algorithms that can work for non-smooth objectives can converge.

---

> > > > > ### Author Response · Authors · 2022-08-10
> > > > > **Reply**
> > > > >
> > > > > Thank you for reconsidering your score! We really appreciate it.
> > > > >
> > > > > Regarding projected gradient: this will have complexity the same as fixed point iterations, since projecting onto the feasible set and inverting the matrix iterates will require O(p^3) time, and computing the gradient will take O(np^2) time - same as fixed points. This is exactly why Frank-Wolfe is so cool for this problem - it’s rank-one updates suits this problem perfectly and lead to an order of the dimension improvement in the runtime of each iteration.
> > > > >
> > > > > Regarding specific contribution: we are quite confident that additional problems of similar structure will turn out that can benefit from this approach.

---

### Author Response · Authors · 2022-08-03
**Revised version**

Dear Reviewers and AC,

Quite embarrassingly, we were not aware until just now that there is an option to revise our submission during the rebuttal period and so we did not plan for it (or allocate time for it).

Nevertheless, we have uploaded a revised supplementary material. In the first appendix A.1 you shall find additional numerical results regarding the original two experimental setups that we had in the original submission. These additional graphs show:
1. the approximation error in spectral norm w.r.t. Tyler's estimator (in log scale), per the comment of Reviewer 6Ljb.
2. the approximation error w.r.t function value in log scale.

We think these new graphs better demonstrate that the variant GAFW can indeed be considerably faster, and that it indeed seems to converge with a linear rate (since the plots are in log scale).

These new plots, as well as additional fixes to the more minor comments will be integrated into our final version.

If there are any more questions we can help with, we would love to do so.

---

### Meta-Review · Area_Chair_Zftb · 2022-08-21

**Recommendation:** Accept
**Confidence:** Certain

**Metareview:**

The scores on this paper were quite spread (and the reviews at times a little imprecise), however looking more closely at the discussion as well as reading the paper myself, I believe this paper should be accepted.

**Award:**

No

---

### Decision · Program_Chairs · 2022-09-14

Accept